

**The evolution of root zone moisture capacities after land**
**use change: a step towards predictions under change?**
**Remko Nijzink[1], Christopher Hutton[2], Ilias Pechlivanidis[4], René Capell[4], Berit**
**Arheimer[4], Jim Freer[2,3], Dawei Han[2], Thorsten Wagener[2,3], Kevin McGuire[5],**
**Hubert Savenije[1], Markus Hrachowitz[1]**
[1] Water Resources Section, Faculty of Civil Engineering and Geosciences, Delft University
of Technology, Stevinweg 1, 2628 CN Delft, The Netherlands
[2] Department of Civil Engineering, University of Bristol, Bristol, UK
[3] Cabot Institute, University of Bristol, BS8 1UJ, Bristol, UK
[4] Swedish Meteorological and Hydrological Institute (SMHI), Norrköping, Sweden
[5] Virginia Water Resources Research Center and Department of Forest Resources and
Environmental Conservation, Virginia Tech, Blacksburg, VA, USA
Correspondence to: R. C. Nijzink (r.c.nijzink@tudelft.nl)





# 1  **Abstract**

The core component of many hydrological systems, the moisture storage capacity available to
vegetation, is impossible to observe directly at the catchment scale and is typically treated as a
calibration parameter or obtained from a priori available soil characteristics combined with
estimates of rooting depth. Often this parameter is considered to remain constant in time. This
is not only conceptually problematic, it is also a potential source of error under the influence
of land use and climate change. In this paper we test the potential of a recently introduced
method to robustly estimate catchment-scale root zone storage capacities exclusively based on
climate data (i.e. rainfall distribution and evaporation) to reproduce the temporal evolution of
root zone storage under change.  Using long-term data from three experimental catchments
that underwent significant land use change, we tested the hypotheses that: (1) root zone
moisture storage capacities are essentially controlled by land cover and climate, (2) root zone
moisture storage capacities are dynamically adapting to changing environmental conditions,
and (3) simple conceptual yet dynamic parametrization, mimicking changes in root zone
storage capacities, can improve a model's skill to reproduce observed hydrological response
dynamics.
It was found that water-balance derived root zone storage capacities were similar to the values
obtained from calibration of four different conceptual hydrological models. A sharp decline in
root zone storage capacity was observed after deforestation, followed by a gradual recovery.
Trend analysis suggested recovery periods between 5 and 13 years after deforestation. In a
proof-of-concept analysis, one of the hydrological models was adapted to allow dynamically
changing root zone storage capacities, following the observed changes due to deforestation.
Although the overall performance of the modified model did not considerably change, it
provided significantly better representations of high flows and peak flows, underlining the
potential of the approach. In 54% of all the evaluated hydrological signatures, considering all
three catchments, improvements were observed when adding a time-variant representation of
the root zone storage to the model.
In summary, it is shown that root zone moisture storage capacities can be highly affected by
deforestation and climatic influences and that a simple method exclusively based on climate-
data can provide robust, catchment-scale estimates of this crucial and dynamic parameter.



## 1   Introduction

Vegetation is a core component of the water cycle, it shapes the partitioning of water fluxes into drainage and evaporation, thereby controlling fundamental processes in ecosystem functioning (Rodriguez-Iturbe, 2000; Laio et al., 2001; Kleidon, 2004), such as flood generation (Donohue et al., 2012), drought dynamics (Seneviratne et al., 2010; Teuling et al., 2013), groundwater recharge (Allison et al., 1990; Jobbágy and Jackson, 2004) and land-atmosphere feedback (Milly and Dunne, 1994; Seneviratne et al., 2013; Cassiani et al., 2015). Besides increasing interception storage available for evaporation (Gerrits et al., 2010), vegetation critically interacts with the hydrological system in a co-evolutionary way by root water uptake for transpiration, towards a dynamic equilibrium with the available soil moisture to avoid water shortage (Donohue et al., 2007; Eagleson, 1978, 1982; Gentine et al., 2012; Liancourt et al., 2012) and related adverse effects on carbon exchange and assimilation rates (Porporato et al., 2004; Seneviratne et al., 2010). By extracting plant available water between field capacity and wilting point, roots create moisture storage volumes within their range of influence. This water holding or root zone storage capacity, $S_R$, in the unsaturated soil is therefore the key component of many hydrological systems (Milly and Dunne, 1994; Rodriguez-Iturbe et al., 2007).

There is increasing theoretical and experimental evidence that vegetation dynamically adapts its root system, and thus $S_R$ , to environmental conditions, balancing between, on the one hand, securing moisture to meet canopy water demand and, on the other hand, minimizing the carbon investment for growth and maintenance of the root system (Brunner et al., 2015; Schymanski et al., 2008; Tron et al., 2015). In other words, the hydrologically active root zone is optimized to guarantee productivity and transpiration of vegetation, given the climatic circumstances (Kleidon, 2004). Several studies already showed the strong influence of climate on this hydrologically active root zone (e.g. Reynolds et al., 2000; Laio et al., 2001; Schenk and Jackson, 2002). Moreover, droughts are often identified as critical situations that can affect ecosystem functioning evolution (e.g. Allen et al., 2010; McDowell et al., 2008; Vose et al.).

In addition to the general adaption to environmental conditions, vegetation has some potential to adapt roots to such periods of water shortage (Sperry et al., 2002; Mencuccini, 2003; Bréda et al., 2006). In the short term, stomatal closure and reduction of leaf area will lead to reduced transpiration. In several case studies for specific plants, it was also shown that plants may





even shrink their roots and reduce soil-root conductivity during droughts, while recovering
after re-wetting (Nobel and Cui, 1992; North and Nobel, 1992). In the longer term, and more
importantly, trees can improve their internal hydraulic system, for example by recovering
damaged xylem or by allocating more biomass for roots (Sperry et al., 2002; Rood et al.,
2003; Bréda et al., 2006). Similarly, Tron et al. (2015) argued that roots follow groundwater
fluctuations, which may lead to increased rooting depths when water tables drop. In addition,
as circumstances change, other species with different water demands may be more in favor in
the competition for resources, as for example shown by Li et al. (2007).
The hydrological functioning of catchments (Black, 1997; Wagener et al., 2007) and thus the
partitioning of water fluxes into evaporation/transpiration and drainage is not only affected by
the continuous adaption of vegetation to changing climatic conditions. Rather, it is well
understood that anthropogenic changes to land cover, such as deforestation, can considerably
alter hydrological regimes. This has been shown historically through many paired watershed
studies (e.g. Bosch and Hewlett, 1982; Andréassian, 2004; Brown et al., 2005; Alila et al.,
2009). These studies found that deforestation often leads to higher seasonal flows and/or an
increased frequency of high flows in streams, while decreasing evaporative fluxes. The time
scales of hydrological recovery after such land use disturbances were shown to be highly
sensitive to climatic conditions and the growth dynamics of the regenerating species (e.g.
Jones and Post, 2004; Brown et al., 2005) .
Although land-use change effects on hydrological functioning are widely acknowledged, it is
less well understood, which parts of the system are affected in which way and over which
time scales. As a consequence, most catchment-scale models were originally not developed to
deal with such changes in the system, but rather for 'stationary' situations (Ehret et al., 2014).
This is valid for both top-down hydrological models, e.g. HBV (Bergström, 1992) or GR4J
(Perrin et al., 2003), and bottom-up models, e.g. MIKE-SHE (Refsgaard and Storm, 1995) or
HydroGeoSphere (Brunner and Simmons, 2012). Several modelling studies have in the past
incorporated temporal effects of land use change to some degree (Andersson and Arheimer,
2001; Bathurst et al., 2004; Brath et al., 2006), but they mostly rely on ad hoc assumptions
about how hydrological parameters are affected (Legesse et al., 2003; Mahe et al., 2005;
Onstad and Jamieson, 1970; Fenicia et al., 2009). More systematic approaches, thus
incorporation the change in the model formulation itself, are rare and have only recently
gained momentum (e.g. Du et al., 2016; Fatichi et al., 2016; Zhang et al., 2016). This is of




critical importance as on-going land use and climate change dictates the need for a better
understanding of their effects on hydrological functioning (Troch et al., 2015) and their
explicit consideration in hydrological models for more reliable predictions under change
(Hrachowitz et al., 2013; Montanari et al., 2013).
As a step towards such an improved understanding and the development of time-dynamic
models, we argue that root zone storage capacity $S_R$ , sometimes also referred to as plant
available water holding capacity, is a core component determining the hydrological response,
and needs to be treated as dynamically evolving parameter in hydrological modelling as a
function of climate and vegetation. Gao et al. (2014) recently demonstrated that catchment-
scale $S_R$ can be robustly estimated exclusively based on long-term water balance
considerations. Wang-Erlandsson et al. (2016) derived global estimates of $S_R$ using remote-
sensing based precipitation and evaporation products, which demonstrated considerable
spatial variability of $S_R$ in response to climatic drivers. In traditional approaches, $S_R$ is
typically determined either by the calibration of a hydrological model (e.g. Seibert and
McDonnell, 2010; Seibert et al., 2010) or based on soil characteristics and sparse estimates of
root depths (e.g. Breuer et al., 2003; Ivanov et al., 2008). This does neither reflect the
dynamic nature of the root system nor does it consider to a sufficient extent the actual
function of the root zone: providing plants with continuous and efficient access to water.  The
main reason for this is that due to the lack of detailed estimates of root depths and their
evolution over time, some average values obtained from literature are typically used. This
leads to the situation that soil porosity often effectively controls $S_R$. Consider, as a thought
experiment, two plants of the same species growing on different soils. They will, with the
same average root depth, then have access to different volumes of water, which will merely
reflect the differences in soil porosity. This is in strong contradiction to the expectation that
these plants would design root systems that provide access to similar water volumes, given
the evidence for efficient carbon investment in root growth (Milly, 1994; Schymanski et al.,
2008; Troch et al., 2009) and posing that plants of the same species have common limits of
operation. This argument is supported by a recent study, in which was shown that water
balance derived estimates of $S_R$ are at least as plausible as soil derived estimates (de Boer-
Euser et al., 2016) in many environments and that the maximum root depth controls
evaporative fluxes and drainage  (Camporese et al., 2015).





Therefore, using water balance based estimates of $S_R$ in several deforested as well as in
untreated reference sites in two experimental forests, we test the hypotheses that (1) the root
zone storage capacity $S_R$ significantly changes after deforestation, (2) changes in $S_R$ can to a
large extent explain post-treatment changes to the hydrological regimes and that (3) a time-
dynamic formulation of $S_R$ can improve the performance of a hydrological model.

## 7   2   Study sites

### 8   2.1   H.J. Andrews Experimental Forest

The H.J. Andrews Experimental Forest is located in Oregon, USA (44.2°N, 122.2°W) and
was established in 1948. The catchments at H.J. Andrews are described in many studies (e.g.
Rothacher, 1965; Dyrness, 1969; Harr et al., 1975; Jones and Grant, 1996; Waichler et al.,
2005) and an overview of the site is presented in Table 1.
Before vegetation removal and at lower elevations the forest generally consisted of 100- to
500-year old coniferous species, such as Douglas-fir (*Pseudotsuga menziesii*), western
hemlock (*Tsuga heterophylla*) and western redcedar (*Thuja plicata*), whereas upper elevations
were characterized by noble fir (*Abies procera*), Pacific silver fir (*Abies amabilis*), Douglas-
fir, and western hemlock. Most of the precipitation falls from November to April (about 80%
of the annual precipitation), whereas the summers are generally drier, leading to signals of
precipitation and potential evaporation that are out of phase. The catchment characteristics of
the watersheds in H.J. Andrews (WS)  are provided in Table 1.
Deforestation of H.J. Andrews WS1 started in August 1962 (Rothacher, 1970). Most of the
timber was removed with skyline yarding. After finishing the logging in October 1966, the
remaining debris was burned and the site was left for natural regrowth. WS2 is the reference
catchment, which was not harvested.

### 25   2.2   Hubbard Brook Experimental Forest

The Hubbard Brook Experimental Forest is a research site established in 1955 and located in
New Hampshire, USA (43.9°N, 71.8°W). The Hubbard Brook experimental catchments are
described in a many publications (e.g. Hornbeck et al., 1970; Hornbeck, 1973; Dahlgren and



Driscoll, 1994; Hornbeck et al., 1997; Likens, 2013). An overview of the site and catchments
used in this study are given in Table1.
Prior to vegetation removal, the forest was dominated by northern hardwood forest composed
of sugar maple (*Acer saccharum*), American beech (*Fagus grandifolia*) and yellow birch
(*Betula alleghaniensis*) with conifer species such as red spruce (*Picea rubens*) and balsam fir
(*Abies balsamea*) occurring at higher elevations and on steeper slopes with shallow soils. The
forest was selectively harvested from 1870 to 1920, damaged by a hurricane in 1938, and is
currently not accumulating biomass (Campbell et al., 2013; Likens, 2013). The annual
precipitation and runoff is less than in H.J. Andrews (Table 1). Precipitation is rather
uniformly spread throughout the year without distinct dry and wet periods, but with snowmelt
dominated peak flows occurring around April and distinct low-flows during the summer
months due to increased evaporation rates (Federer et al., 1990). Vegetation removal occurred
in the catchment of WS2 between 1965-1968 and in WS5 between 1983-1984. Hubbard
Brook WS3 is the undisturbed reference catchment.
Hubbard Brook WS2 was completely deforested in November and December 1965 (Likens et
al., 1970). To minimize disturbance, no roads were constructed and all timber was left in the
catchment. On June 23, 1966, herbicides were sprayed from a helicopter to prevent regrowth.
Additional herbicides were sprayed in the summers of 1967 and 1968 from the ground.
In Hubbard Brook WS5, all trees were removed between October 18, 1983 and May 21, 1984,
except for a 2 ha buffer near an adjacent reference catchment (Hornbeck et al., 1997). WS5
was harvested as a whole-tree mechanical clearcut with removal of 93% of the above-ground
biomass (Hornbeck et al., 1997; Martin et al., 2000); thus, including smaller branches and
debris. Approximately 12% of the catchment area was developed as the skid trail network.
Afterwards, no treatment was applied and the site was left for regrowth.
**3   Methodology**
To assure reproducibility and repeatability, the executional steps in the experiment were
defined in a detailed protocol, following Ceola et al. (2015), which is provided as
supplementary material in Section S1.



### 3.1 Water balance-derived root zone moisture capacities $S_R$

The root zone moisture storage capacities $S_R$ and their change over time were determined according to the methods suggested by Gao et al. (2014), de Boer-Euser et al. (2016) and Wang-Erlandsson et al. (2016). Briefly, the long-term water balance provides information on actual mean transpiration. In a first step, the interception capacity has to be assumed, in order to determine the effective precipitation $P_e$ [L $T^{-1}$], following the water balance equation for interception storage:

$$\frac{dS_i}{dt} = P - E_i - P_e$$

$$(1)$$

With $S_i$ [L] interception storage, $P$ the precipitation [L $T^{-1}$], $E_i$ the interception evaporation [L $T^{-1}$]. This is solved with the constitutive relations:

$$E_i = \begin{cases} E_p & if\ E_p dt < S_i \\ \frac{S_i}{dt} & if\ E_p dt \geq S_i \end{cases}$$

$$(2)$$

$$P_e = \begin{cases} 0 & if\ S_i \leq I_{max} \\ \frac{S_i - I_{max}}{dt} & if\ S_i > I_{max} \end{cases}$$

$$(3)$$

With, additionally, $E_p$ the potential evaporation [L $T^{-1}$] and $I_{max}$ [L] the interception capacity. Nevertheless, $I_{max}$ will also be affected by land use change. This was addressed by introducing the three parameters $I_{max,eq}$ (long-term equilibrium interception capacity) [L], $I_{max,change}$ (post-treatment interception capacity) [L] and $T_r$ (recovery time) [T], leading to a time-dynamic formulation of $I_{max}$:

$$I_{max} = \begin{cases} I_{max,eq} & for\ t < t_{change},\, t > t_{change,end} + T_r \\ I_{max,eq} - \frac{I_{max,eq} - I_{max,change}}{t_{change,end} - t_{change,start}}\left(t - t_{change,start}\right) & for\ t_{change,start} < t < t_{change,end} \\ I_{max,change} + \frac{I_{max,eq} - I_{max,change}}{T_r}\left(t - t_{change,end}\right) & for\ t_{change,end} < t < t_{change,end} + T_r \end{cases}$$

$$(4)$$

with $t_{change,start}$ the time that deforestation started and $t_{start,end}$ the time deforestation finished.





Following a Monte-Carlo sampling approach, upper and lower bounds of $E_i$ were then
estimated based on 1000 random samples of these parameters, eventually leading to upper and
lower bounds for $P_e$. The interception capacity was assumed to increase after deforestation for
Hubbard Brook WS2, as the debris was left at the site. For Hubbard Brook WS5 and HJ
Andrews WS1 the interception capacity was assumed to decrease after deforestation, as here
the debris was respectively burned and removed. Furthermore, in the absence of more detailed
information, it was assumed that the interception capacities changed linearly during
deforestation towards $I_{max,change}$ and linearly recovered to $I_{max}$ over the period $T_r$ as well. See
Table 2 for the applied parameter ranges.
Hereafter, the long term mean transpiration can be estimated with the remaining components
of the long term water balance, assuming no additional gains/losses, storage changes and/or
data errors:
$$\overline{E_t} = \overline{P_e} - \overline{Q}, \tag{5}$$
where $E_t$ [L T$^{-1}$] is the long-term mean actual transpiration, $P_e$ [L T$^{-1}$] is the long-term mean
effective precipitation and Q [L T$^{-1}$] is the long-term mean catchment runoff. Taking into
account seasonality, the actual mean transpiration is scaled with the ratio of long-term mean
daily potential evaporation $E_p$ over the mean annual potential evaporation $E_p$:
$$E_t(t) = \frac{E_p(t)}{E_p} * \overline{E_t} \tag{6}$$
Based on this, the cumulative deficit between actual transpiration and precipitation over time
can be estimated by means of an 'infinite-reservoir'. In other words, the cumulative sum of
daily water deficits, i.e. evaporation minus precipitation, is calculated between $T_0$, which is
the time the deficit equals zero, and $T_1$, which is the time the total deficit returned to zero. The
maximum deficit of this period then represents the volume of water that needs to be stored to
provide vegetation continuous access to water throughout that time:
$$S_R = \max \int_{T0}^{T1} (E_t - P_e) \, dt, \tag{7}$$
where $S_R$ [L] is the maximum root zone storage capacity over the time period between $T_0$ and
$T_1$. See also Figure 1 for a graphical example of the calculation for the Hubbard Brook



catchment for one specific realization of the parameter sampling. The $S_{R,20yr}$ for drought
return periods of 20 years was estimated using the Gumbel extreme value distribution
(Gumbel, 1941) as previous work suggested that vegetation designs $S_R$ to satisfy deficits
caused by dry periods with return periods of approximately 10-20 years (Gao et al., 2014; de
Boer-Euser et al., 2016). Thus, the yearly values of $S_R$, as obtained by equation 6, were fitted
to the extreme value distribution of Gumbel, and subsequently, the $S_{R,20yr}$ was determined.
For the study catchments that experienced logging and subsequent reforestation, it was
assumed that the root system converges towards a dynamic equilibrium approximately 10
years after reforestation. Thus, the equilibrium $S_{R,20yr}$ was estimated using only data over a
period that started at least 10 years after the treatment. For the growing root systems during
the years after reforesting, the storage capacity does not yet reach its dynamic equilibrium
$S_{R,20yr}$. Instead of determining an equilibrium value, the maximum occurring deficit for each
year was in that case considered as the maximum demand and thus as the maximum required
storage $S_{R,1yr}$ for that year. To make these yearly estimates, the mean transpiration was
determined in a similar fashion as stated by Equation 2. However, the assumption of no
storage change may not be valid for 1-year periods. In a trade-off, the mean transpiration was
determined based on the 2-year water balance, thus assuming no storage change over these
years.
The deficits in the months October-April are highly affected by snowfall, as estimates of the
effective precipitation are estimated without accounting for snow, leading to soil moisture
changes that spread out over an unknown longer period due to the melt process. Therefore, to
avoid this influence of snow, only deficits as defined by Equation 5, in the period of May –
September are taken into consideration, which is also the period where deficits are caused by
relatively low rainfall precipitation and high transpiration rates, thus causing soil moisture
depletion and drought stress for the vegetation, and in turn, shaping the root zone.
**3.2   Model-derived root zone storage capacity $S_{u,max}$**
The water balance derived equilibrium $S_{R,20yr}$ as well as the dynamically changing $S_{R,1yr}$ that
reflects regrowth patterns in the years after treatment were compared with estimates of the
calibrated parameter $S_{u,max}$, which represents the mean catchment root zone storage capacity
in lumped conceptual hydrological models. Due to the lack of direct observations of the
changes in the root zone storage capacity, this comparison was used to investigate whether the





estimates of the root zone storage capacity $S_{R,1yr}$, and their sensitivity to land use change as
well as their effect on hydrological functioning, can provide similar results as the model-
based root zone storage. Model-based estimates of root zone storage capacity may be highly
influenced by model formulations and parameterizations. Therefore, four different
hydrological models were used to derive the parameter of $S_{u,max}$ in order to obtain a set of
different estimates of the catchment scale root zone storage capacity. The major features of
the model routines for root-zone moisture tested here are briefly summarized below and
detailed descriptions including the relevant equations are provided as supplementary material
(Section S2).

### 3.2.1 FLEX

A FLEX-based model (Fenicia et al., 2008) was applied in a lumped way to the catchments. It
consists of five storage components. First, a snow routine has to be run before the
precipitation enters the interception reservoir. Here, water evaporates at potential rates or,
when exceeding a threshold, continues to the soil moisture reservoir. The soil moisture
routine is modelled in a similar way as the Xinanjiang model (Zhao, 1992). Briefly, it
contains a distribution function that determines the fraction of the catchment where the
storage deficit in the root zone is satisfied and that is therefore connected to the stream and
generating storm runoff. From the soil moisture reservoir, water can further percolate down to
the groundwater or leave the reservoir through transpiration.
Water that cannot be stored in the soil moisture storage then is split into preferential
percolation to the groundwater and runoff generating fluxes that enter a fast reservoir, which
represents fast responding system components such as shallow subsurface and overland flow.

### 3.2.2 HYPE

The HYPE model (Lindström et al., 2010) estimates soil moisture for Hydrological Response
Units (HRU), which is the finest calculation unit in this catchment model. Each HRU consist
of a unique combination of soil and land-use classes with assigned soil depth. Water input is
estimated from precipitation after interception and a snow module at the catchment scale,
after which the water enters the three defined soil layers in each HRU. Evaporation and
transpiration takes place from the first two layers and fast surface runoff is produced when
these layers are fully saturated or when rainfall rates exceeds the maximum infiltration
capacities. Water can move between the layers through percolation or laterally via fast flow





pathways. The catchment can also receive input of lateral flow from upper sub-catchments.
The groundwater table is fluctuating between the soil layers with the lowest soil layer
normally reflecting the base flow component in the hydrograph. The water balance of each
HRU is calculated independently and the runoff is then aggregated in a local stream with
routing before entering the main stream.

### 3.2.3 TUW

The TUW model (Parajka et al., 2007) is a conceptual model with a structure similar to that of
HBV (Bergström, 1976). After a snow module, water enters a soil moisture routine. From this
soil moisture routine, water is partitioned into runoff generating fluxes and transpiration. The
runoff generating fluxes percolate into two series of reservoirs. A fast responding reservoir
with overflow outlet represents shallow subsurface and overland flow, while the slower
responding reservoir represents the groundwater.

### 3.2.4 HYMOD

HYMOD (Boyle, 2001) is similar to the applied model structure for FLEX, besides that the
interception module and percolation from soil moisture to the groundwater are missing.
Nevertheless, the model accounts similarly for the partitioning of transpiration and runoff
generation in a soil moisture routine. The runoff generating fluxes are then divided over a
slow reservoir, representing groundwater, and a fast reservoir, representing the fast processes.

### 3.3 Model calibration

Each model was calibrated using a Monte-Carlo strategy within consecutive two year
windows in order to obtain a time series of root zone moisture capacities $S_{u,max}$. The Kling-
Gupta efficiency for flows (Gupta et al., 2009), the Kling-Gupta efficiency for the logarithm
of the flows and the Volume Error (Criss and Winston, 2008) were simultaneously used as
objective functions in a multi-objective calibration approach to evaluate the model
performance for each window. These were selected in order to obtain rather balanced
solutions that enable a sufficient representation of peak flows, low flows and the water
balance. The unweighted Euclidian Distance $D_E$ of the three objective functions served as an
informal measure to obtain these balanced solutions (e.g. Hrachowitz et al., 2014; Schoups et
al., 2005):



$$L(\theta) = 1 - \sqrt{(1 - E_{KG})^2 + (1 - E_{logKG})^2 + (1 - E_{VE})^2} \qquad (8)$$

where $L(\theta)$ is the conditional probability for parameter set $\theta$ [-], $E_{KG}$ the Kling-Gupta
efficiency [-], $E_{logKG}$ the Kling-Gupta efficiency for the log of the flows [-], and $E_{VE}$ the
volume error [-].
Eventually, a weighing method based on the GLUE-approach of Freer et al. (1996) was
applied. To estimate posterior parameter distributions all solutions with Euclidian Distances
smaller than 1 were maintained as feasible. The posterior distributions were then determined
with the Bayes rule  (cf. Freer et al., 1996):
$$L_2(\theta) = L(\theta)^n * L_0(\theta)/C \qquad (8)$$
where $L_0(\theta)$ is the uninformed prior parameter distribution [-], $L_2(\theta)$ the posterior conditional
probability [-] and C a normalizing constant [-]. 5/95[th] model uncertainty intervals were then
constructed based on the posterior conditional probabilities.
**3.4   Trend analysis**
To test if $S_{R,1yr}$ significantly changes following de- and subsequent reforestation, which would
also indicate shifts in distinct hydrological regimes, a trend analysis, as suggested by Allen et
al. (1998), was applied to the $S_{R,1yr}$ values obtained from the water balance-based method. As
the sampling of interception capacities (Eq. 4) leads to $S_{R,1yr}$ values for each point in time,
which are all equally likely in absence of any knowledge, the mean of this range was assumed
as an approximation of the time-dynamic character of $S_{R,1yr}$.
Briefly, a linear regression between the full series of the cumulative sums of $S_{R,1yr}$ in the
deforested catchment and the unaffected control catchment is established and the residuals
and the cumulative residuals are plotted in time. A 95%-confidence ellipse is then constructed
from the residuals:
$$X = \frac{n}{2}\cos(\alpha) \qquad (9)$$


$$Y = \frac{n}{\sqrt{n-1}} Z_{p95}\, \sigma_r\, \sin(\alpha)$$

1      (10)

where X presents the x-coordinates of the ellipse [T], Y represents the y-coordinates of the
ellipse [L], n is the length of the time series [T], $\alpha$ is the angle defining the ellipse (0 - $2\pi$)
between the diagonal of the ellipse and the x-axis [-], $Z_{95}$ is the value belonging to a
probability of 95% of the standard student t-distribution [-] and $\sigma_r$ is the standard deviation of
the residuals (assuming a normal distribution) [L].
When the cumulative sums of the residuals plot outside the 95%-confidence interval defined
by the ellipse, the null-hypothesis that the time series are homogeneous is rejected. In that
case, the residuals from this linear regression where residual values change from either solely
increasing to decreasing or vice versa, can then be used to identify different sub-periods in
time.
Thus, in a second step, for each identified sub-period a new regression, with new (cumulative)
residuals, can be used to check homogeneity for these sub-periods. In a similar way as before,
when the cumulative residuals of these sub-periods now plot within the accompanying newly
created 95%-confidence ellipse, the two series are homogeneous for these sub-periods. In
other words, the two time series show a consistent behavior over this particular period.
## 3.5   Model with time-dynamic formulation of $S_{u,max}$
In a last step, the FLEX model was reformulated to allow for a time-dynamic representation
of the parameter $S_{u,max}$, reflecting the root zone storage capacity.
As a reference, the long-term water balance derived root zone storage capacity $S_{R,20yr}$ was
used as a static formulation of $S_{u,max}$ in the model, and thus kept constant in time. The
remaining parameters were calibrated using the calibration strategy outlined above over a
period starting with the treatment in the individual catchments until at least 15 years after the
end of the treatment. This was done to focus on the period under change (i.e. vegetation
removal and recovery), during which the differences between static and dynamic formulations
of $S_{u,max}$ are assumed to be most pronounced.
To test the effect of a dynamic formulation of $S_{u,max}$ as a function of forest regrowth, the
calibration was run with a series temporally evolving root zone storage capacities, similar to



formulations of leaf area index and overstore height for the DHSVM model by Waichler et al.
(2005). The time-dynamic series of $S_{u,max}$ were obtained from a relatively simple growth
function, the Weibull function (Weibull, 1951):
$$S_{u,max}(t) = S_{R,20y} \left(1 - e^{-a*t^b}\right),$$  (11)
where $S_{u,max}(t)$ is the root zone storage capacity $t$ time steps after reforestation [L], $S_{R,20yr}$ is
the equilibrium value [L], and $a$ [T$^{-1}$] and $b$ [-] are shape parameters. In the absence of more
information, this equation was selected as a first, simple way of incorporating the time-
dynamic character of the root zone storage capacity in a conceptual hydrological model. In
this way, root growth is exclusively determined dependent on time, whereas the shape-
parameters $a$ and $b$ merely implicitly reflect the influence of other factors, such as climatic
forcing in a lumped way. These parameters were estimated based on qualitative judgement so
that $S_{u,max}(t)$ coincides well with the suite of $S_{R1yr}$ values after logging. This approach was
followed to filter out the short term fluctuations in the $S_{R1yr}$ values, which is not warranted by
this equation. In addition, it should be noted that this rather simple approach is merely meant
as a proof-of-concept for a dynamic formulation of $S_{u,max}$.
In addition, the remaining parameter directly related to vegetation, the interception capacity
(*Imax*), was also assigned a time-dynamic formulation. Here, the shape of the growth function
was assumed fixed (i.e. growth parameters $a$ and $b$ were fixed to values of 0.001 [day$^{-1}$] and 1
[-]) loosely based on the posterior ranges of the window calibrations. This growth function
was used to ensure the degrees of freedom for both the time-variant and the time-invariant
models, leaving the equilibrium value of the interception capacity as the only free calibration
parameter for this process. Note that the empirically parameterized growth functions can be
readily extended and/or replaced by more mechanistic, process-based descriptions of
vegetation growth if warranted by the available data and was here merely used to test the
effect of considering changes in vegetation on the skill of models to reproduce hydrological
response dynamics.
To assess the performance of the dynamic model compared to the time-invariant formulation,
beyond the calibration objective functions, model skill in reproducing 28 hydrological
signatures was evaluated (Sivapalan et al., 2003). Even though the signatures are not always
fully independent of each other, this larger set of measures allows a more complete evaluation





of the model skill as, ideally, the model should be able to perfectly and simultaneously
reproduce each signature. An overview of the signatures is given in Table 2. The results of the
comparison were quantified on the basis of the probability of improvement for each signature
(Nijzink et al., 2016):
$$P_{I,S} = P(S_{dyn} > S_{stat}) = \sum_{i=1}^{n} P(S_{dyn} > S_{stat} \mid S_{dyn} = r_i) P(S_{dyn} = r_i) \tag{12}$$
where $S_{dyn}$ and $S_{stat}$ are the distributions of the signature performance metrics of the dynamic
and static model, respectively, for the set of all feasible solutions retained from calibration, $r_i$
is a single realization from the distribution of $S_{dyn}$ and $n$ is the total number of realizations of
the $S_{dyn}$ distribution. For $P_{I,S} > 0.5$ it is then more likely that the dynamic model outperforms
the static model with respect to the signature under consideration, and vice versa for $P_{I,S} < 0.5$.
The signature performance metrics that were used are the relative error for single-valued
signatures and the Nash-Sutcliffe efficiency (Nash and Sutcliffe, 1970) for signatures that
represent a time series.
In addition, as a more quantitative measure, the Ranked Probability Score, giving information
on the magnitude of model improvement or deterioration, was calculated (Wilks, 2005):

$$S_{RP} = \frac{1}{M-1} \sum_{m=1}^{M} \left[ \left( \sum_{k=1}^{m} p_k \right) - \left( \sum_{k=1}^{m} o_k \right) \right]^2$$

16  (13)

where M is the number of feasible solutions, $p_k$ the probability of a certain signature
performance to occur and $o_k$ the probability of the observation to occur (either 1 or 0, as there
is only a single observation). Briefly, the $S_{RP}$ represents the area enclosed between the
cumulative probability distribution obtained by model results and the cumulative probability
distribution of the observations. Thus, when modelled and observed cumulative probabilities
are identical, the enclosed area goes to zero. Therefore, the difference between the $S_{RP}$ for the
feasible set of solutions for the time-variant and time-invariant model formulation was used in
the comparison, identifying which model is quantitatively closer to the observation.





## 4 Results and Discussion

### 4.1 Deforestation and changes in hydrological response dynamics

We found that the three deforested catchments in the two research forests show generally similar response dynamics after the logging of the catchments (Fig.2). This supports the findings from previous studies of these catchments (Andréassian, 2004; Bosch and Hewlett, 1982; Hornbeck et al., 1997; Rothacher et al., 1967). More specifically, it was found that the observed annual runoff coefficients for HJ Andrews WS1 and Hubbard Brook WS2 (Fig. 2a,b) change after logging of the catchments, also in comparison with the reference watersheds. Right after deforestation, runoff coefficients increase, but are followed by a gradual decrease. This change in runoff behavior points towards shifts in the yearly sums of transpiration, which can, except for climatic variation, be linked to the regrowth of vegetation that takes place at a similar pace as the changes in hydrological dynamics. This coincidence of regrowth dynamics and evolution of runoff coefficients was not only noticed by Hornbeck et al. (2014) for the Hubbard Brook, but was also previously acknowledged for example by Swift and Swank (1981) in the Coweeta experiment or Kuczera (1987) for eucalypt regrowth after forest fires. The key role of vegetation in this partitioning between runoff and transpiration (Donohue et al., 2012), or more specifically root zones (Gentine et al., 2012), necessarily leads to a change in runoff coefficients when vegetation is removed. Similarly, Gao et al. (2014) found a strong correlation between root zone storage capacities and runoff coefficients in more than 300 US catchments, which lends further support to the hypothesis that root zone storage capacities may have decreased in deforested catchments right after removal of the vegetation.

The annual autocorrelation coefficients with a 1-day lag time are generally lower after logging than in the years before the change, which can be seen in particular from Figures 2e and 2f as here a long pre-treatment time series record is available. Nevertheless, the climatic influence cannot be ignored here, as the reference watershed shows a similar pattern. Only for Hubbard Brook WS5 (Fig. 2f), the autocorrelation shows reduced values in the first years after logging. Thus, the flows at any time $t+1$ are less dependent on the flows at $t$, which points towards less memory and thus less storage in the system (i.e. reduced $S_R$), leading to increased peak flows, similar to the reports of, for example, Patric and Reinhart (1971) for one of the Fernow experiments.





The declining limb density for HJ Andrews WS1 (Fig. 2g) shows increased values right after
deforestation, whereas longer after deforestation the values seem to plot closer to the values
obtained from the reference watershed. This indicates that for the same number of peaks less
time was needed for the recession in the hydrograph in the early years after logging. In
contrast, the rising limb density shows increased values during and right after deforestation
for Hubbard Brook WS2 and WS5 (Fig 2k-2l), compared to the reference watershed. Here,
less time was needed for the rising part of the hydrograph in the more early years after
logging. Thus, the recession seems to be affected in HJ Andrews WS1, whereas the Hubbard
Brook watersheds exhibits a quicker rise of the hydrograph.
Eventually, the flow duration curves, as shown in Figures 2m-2o, indicate a higher variability
of flows, as the years following deforestation plot with an increased steepness of the flow
duration curve, i.e. a higher flashiness. This increased flashiness of the catchments after
deforestation can also be noted from the hydrographs shown in Figure 3. The peaks in the
hydrographs are generally higher, and the flows return faster to the baseflow values in the
years right after deforestation than some years l later after some forest regrowth, all with
similar values for the yearly sums of precipitation and potential evaporation.
**4.2   Temporal evolution of $S_R$ and $S_{u,max}$**
The observed changes in the hydrological response of the study catchments (as discussed
above) were also clearly reflected in the temporal evolution of the root zone storage capacities
as described by the catchment models (Fig. 4). The models all exhibited Kling-Gupta
efficiencies ranging between 0.5 and 0.8 and Kling-Gupta efficiencies of the log of the flows
between 0.2 and 0.8 (see the supplementary material Figures S5-7, with all posterior
parameter distributions in Figures S9-S26). Comparing the water balance and model-derived
estimates of root zone storage capacity $S_R$ and $S_{u,max}$, respectively, then showed that they
exhibit very similar patterns in the study catchments. In general, root zone storage capacities
sharply decreased after deforestation and, when regrowth occurred, gradually recovered
towards a dynamic equilibrium of climate and vegetation, whereas the reference catchments
of HJ Andrews WS2 and Hubbard Brook WS3 showed a rather constant signal over the full
period (see the supplementary material Figure S8). This in agreement with Mahe et al. (2005),



who found in a modelling exercise that water holding capacities needed to be lowered after a
reduction in vegetation.
The HJ Andrews WS1 shows the clearest signal when looking at the water balance derived
$S_R$, as can be seen by the green shaded area in Figure 4a. Before deforestation, the root zone
storage capacity $S_{R,1yr}$ was found to be around 400mm. In spite of the high annual
precipitation volumes, such comparatively high $S_{R,1yr}$ is plausible given the marked
seasonality of the precipitation in the Mediterranean climate (Koeppen-Geiger class Csb) and
the approximately 6 months phase shift between precipitation and potential evaporation peaks
in the study catchment, which dictates that the storage capacities need to be large enough to
store precipitation falling mostly during winter throughout the extended dry periods with
higher energy supply throughout the rest of the year (Gao et al., 2014). During deforestation,
the $S_{R,1yr}$ required to provide the remaining vegetation with sufficient and continuous access to
water decreased from around 400 mm to 200 mm. For the first 4-6 years after deforestation
the $S_{R,1yr}$ increased again, reflecting the increased water demand of  vegetation with the
regrowth of the forest.
The four models show a similar pronounced decrease of the calibrated, feasible set of $S_{u,max}$
during deforestation and a subsequent gradual increase over the first years after deforestation.
The model concepts, thus our assumptions about nature, can therefore only account for the
changes in hydrological response dynamics of a catchment, when calibrated in a window
calibration approach with different parameterizations for each time frame. The absolute
values of $S_{u,max}$ obtained from the most parsimonious HYMOD and FLEX models (both 8
free calibration parameters) show a somewhat higher similarity to $S_{R,1yr}$ and its temporal
evolution than the values from the other two models. In spite of similar general patterns in
$S_{u,max}$, the higher number of parameters in TUW (i.e. 15) result, due to compensation effects
between individual parameters, in wider uncertainty bounds which are less sensitive to
change. It was also observed that in particular TUW overestimates $S_{u,max}$ compared to $S_{R,1yr}$,
which is caused by the absence of an interception reservoir, leading to a root zone that has to
satisfy not only transpiration but all evaporative fluxes.
It was observed that in the period 1971- 1978 $S_{R,1yr}$ slowly decreased again in HJ Andrews.
This pattern indicates that the storage demand in these years was lower as more rainfall
reduced the need for storage in the system, which can be seen from the rainfall chart on top of
Figure 4a. This reduced demand for storage could potentially indicate a contracting root



system during that period, as an effort of vegetation to optimize its subsurface energy and
carbon allocation for root maintenance in a trade-off for increased above-surface growth.
However, this conclusion is at this point not warranted by the available data and it can also be
argued that the system is in a state of over-capacity for that period, still maintaining the root
systems for the dryer years to come. The hydrograph for the years 1978-1979 (Figure 5)
rather support the latter. Even though the FLEX model calibrated for this period tended
towards larger values of $S_{u,max}$ (Figure 4a), still the modelled peaks are relatively high
compared to the observed peaks. This suggests that the model requires a higher buffer in the
root zone to reduce the peak flows rather than that root zones should have contracted in this
time of reduced need. Thus, from 1980 and onwards the system can rather easily survive the
period of growing demand caused by the relatively dry and warm years.
Hubbard Brook WS2 exhibits a similarly clear decrease in root zone storage capacity as a
response to deforestation, as shown in Figure 4b. The water balance-based $S_{R,1yr}$ estimates
approach values of zero during and right after deforestation. In these years the catchment was
treated with herbicides, removing effectively any vegetation, thereby minimizing
transpiration. Low $S_{R,1yr}$ values are highly plausible in this catchment because the relatively
humid climate and the absence of pronounced rainfall seasonality strongly reduces storage
requirements (Gao et al., 2014). In this catchment a more gradual regrowth pattern occurred,
which continued after logging started in 1966 until around 1983. However, the marked
increase in $S_{R,1yr}$ at that time rather points towards an exceptional year, in terms of
climatological factors, than a sudden expansion of the root zone. It can also be observed from
Figure 3a that the runoff coefficient was relatively low for 1985, suggesting either increased
evaporation or a storage change. It can be argued, that a combination of a relatively long
period of low rainfall amounts and high potential evaporation, as can be noted by the
relatively high mean annual potential evaporation on top of Figure 4b, led to a high demand in
1985. Parts of the vegetation may not have survived these high-demand conditions due to
insufficient access to water, which in turn can explain the dip in $S_{R,1yr}$ for the following year,
which is in agreement with reduced growth rates of trees after droughts as observed by for
example Bréda et al. (2006).
The hydrographs of 1984-1985 (Figure 6a) and 1986-1987 (Figure 6b) also show that July-
August 1985 was exceptionally dry, whereas the next year in August 1986 the catchment
seems to have increased peak flows. This either points towards an actual low storage capacity





due to contraction of the roots during the dry summer or a low need of the system to use the
existing capacity, for instance to recover other vital aspects of the system.
Generally, the models applied in Hubbard Brook WS2 show similar behavior as in the HJ
Andrews catchment. The calibrated $S_{u,max}$ clearly follows the temporal pattern of $S_{R,1yr}$,
reflecting the pronounced effects of de- and reforestation. It can, however, also be observed
that the absolute values of $S_{u,max}$ exceed the $S_{R,1yr}$ estimates. While FLEX on balance exhibits
the closest resemblance between the two values, in particular the TUW model exhibits wide
uncertainty bounds with elevated $S_{u,max}$ values. Besides the role of interception evaporation,
which is only explicitly accounted for in FLEX,  the results are also linked to the fact that the
humid climatic conditions with little seasonality reduces the importance of the model
parameter $S_{u,max}$, and makes it thereby more difficult to identify by calibration. The parameter
is most important for lengthy dry periods  when vegetation needs enough storage to ensure
continuous access to water.
The temporal variation in $S_R$ in Hubbard Brook WS5 does not show such a distinct signal as
in the other two study catchments (Figure 4c). Here the forest was removed in a whole-tree
harvest in winter '83-'84 followed by natural regrowth. The summers of 1984 and 1985 were
very dry summers, as also reflected by the high values of $S_{R,1yr}$. The young system had
already developed enough roots before these dry periods to have access to a sufficiently large
water volume to survive this summer. This is plausible, as the period of the highest deficit
occurred in mid-July and lasted until approximately the end of September, thus long after the
growing season, allowing enough time for an initial growth and development of young roots
from April until mid-July. In addition, the composition of the new forest differed from the old
forest with more pin cherry (*Prunus pensylvanica*) and paper birch (*Betula papyrifera*). This
supports the statements of a quick regeneration as these species have a high growth rate and
reach canopy closure in a few years. Furthermore, the forest was not treated with either
herbicides (Hubbard Brook WS2) or burned (HJ Andrews WS1), leaving enough low shrubs
and herbs to maintain some level of transpiration (Hughes and Fahey, 1991; Martin, 1988). It
can thus be argued, similar to Li et al. (2007), that the remaining vegetation experienced less
competition and could increase root water uptake efficiency and transpiration per unit leaf
area. This is in agreement with Hughes and Fahey (1991), who also stated that several species
benefited from the removal of canopies and newly available resources in this catchment.
Lastly, several other authors related the absence of a clear change in hydrological dynamics to



the severe soil disturbance in this catchment (Hornbeck et al., 1997; Johnson et al., 1991).
These disturbances lead to extra compaction, whereas at the same time species were changing,
effectively masking any changes in runoff dynamics.

## 4.3   Process understanding - trend analysis and change in hydrological regimes

The trend analysis for water-balance derived values of $S_{R,1yr}$ suggests that for all three study
catchments significantly different hydrological regimes in time can be identified before and
after deforestation, linked to changes in $S_{R,1yr}$ (Fig. 7). For all three catchments, the
cumulative residuals plot outside the 95%-confidence ellipse, indicating that the time series
obtained in the control catchments and the deforested catchments are not homogeneous
(Figures 7g-7i).
Rather obvious break points can be identified in the residuals plots for the catchments HJ
Andrews WS1 and Hubbard Brook WS2 (Fig. 7d-7e). Splitting up the $S_{R,1yr}$ time series
according to these break points into the periods before deforestation, deforestation  and
recovery resulted in three individually homogenous time series that are significantly different
from each other, indicating switches in the hydrological regimes. The results shown in Figure
4 indicate that these catchments had a rather stable root zone storage capacity during
deforestation. Hence, recovery and deforestation balanced each other, leading to a temporary
equilibrium. The recovery signal then becomes more dominant in the years after
deforestation. The third homogenous period suggests that the root zone storage capacity
reached a dynamic equilibrium without any further systematic changes. This can be
interpreted in the way that in the HJ Andrews WS1 hydrological recovery after deforestation
due to the recovery of the root zone store capacity took about 6-9 years (Fig. 7p), while
Hubbard Brook WS2 required 10-13 years for hydrological  recovery (Fig. 7q). This strongly
supports the results of Hornbeck et al. (2014), who reported changes in water yield for WS2
for up to year 12 after deforestation.
The identification of different periods is less obvious for Hubbard Brook WS5, but the two
time series of control catchment and treated catchment are significantly different (see the
cumulative residuals in Figure 7i). Nevertheless, the most obvious break point in residuals can
be found in 1989 (Figure 7f).  In addition, it can be noted that turning points also exist in 1983
and 1985. These years can be used to split the time series into four groups (leading to the





periods of 1964-1982, 1983-1985, 1986-1989 and 1990-2009 for further analysis). The
cumulative residuals from the new regressions, based on the grouping, plot within the
confidence bounds again, and show a period with deforestation (1983-1985) and recovery
(1986-1989). Mou et al. (1993) reported similar findings with the highest biomass
accumulation in 1986 and 1988, and slower vegetation growth in the early years. Therefore,
full recovery took 5-6 years in Hubbard Brook WS5.
The above results do in general suggest similar recovery periods for forest systems as reported
in earlier studies, such as Brown et al. (2005) or Hornbeck et al. (2014), who found that
catchments reach a new equilibrium with a similar timescale as reported here with the direct
link to the parameter describing the catchment-scale root zone storage capacity. The
timescales are also in agreement with regression models to predict water yield after logging of
Douglass (1983), who assumed a duration of water yield increases of 12 years for coniferous
catchments.   The timescales found here are around 10 years (here 5-13 years for the
catchments under consideration), but will probably depend on climatic factors and vegetation
type.
**4.4   Time-variant model formulation**
The adjusted model routine for FLEX, which uses a dynamic time series of $S_{u,max}$, generated
with the Weibull growth function (Eq.11), resulted in a rather small impact on the overall
model performance in terms of  the calibration objective function values (Figure 8b, 8d, 8f)
compared to the time-invariant formulation of the model. The strongest improvements for
calibration were observed for the dynamic formulation of FLEX for HJ Andrews WS1 and
Hubbard Brook WS2 (Figures 8b and 8d), which reflects the rather clear signal from
deforestation in these catchments.
Evaluating a set of hydrological signatures suggests that the dynamic formulation of $S_{u,max}$
allows the model to have a higher probability to better reproduce most of the signatures tested
here (54% of all signatures in the three catchments) as shown in Figure 9a. A similar pattern
is obtained for the more quantitative $S_{RP}$ (Figure 9b), where in 52% of the cases improvements
are observed. Most signatures for HJ Andrews WS1 show a high probability of improvement,
with a maximum $P_{I,S}$ =0.69 (for $Q_{95,winter}$) and an average $P_{I,S}$ = 0.55. Considering the large
difference between the deforested situation and the new equilibrium situation of about 200
mm, this supports the hypothesis that here a time-variant formulation of $S_{u,max}$ does provide





means for an improved process representation and, thus, hydrological signatures. Here,
improvements are observed especially in the high flows in summer ($Q_{5,summer}$, $Q_{50,summer}$) and
peak flows (e.g. Peaks, Peaks$_{summer}$, Peaks$_{winter}$), that illustrates that the root zone storage
affects mostly the fast responding components of the system as also suggested previously
(e.g. de Boer-Euser et al., 2016; Euser et al., 2015; Oudin et al., 2004), by providing a buffer
to storm response. In addition, a dynamic formulation of $S_{u,max}$ permits a more plausible
representation of the variability in land-atmosphere exchange following land use change,
which is a critical input to climate models (Entekhabi et al., 1996; Seneviratne et al., 2010).
Fulfilling its function as a storage reservoir for plant available water, modelled transpiration is
significantly reduced post-deforestation, which in turn results in increased runoff coefficients
(cf. Gao et al., 2014), which have been frequently reported for post-deforestation periods by
earlier studies (e.g. Hornbeck et al., 2014; Rothacher, 1970; Swift and Swank, 1981) .
At Hubbard Brook WS2 a more variable pattern is shown in the ability of the model to
reproduce the hydrological signatures. It is interesting to note that the low flows ($Q_{95}$
,$Q_{95,summer}$, $Q_{50,summer}$) improve, opposed to the expectation raised by the argumentation for HJ
Andrews WS1 that peak flows and high flows should improve. In this case, the peaks are too
high for the time-dynamic model. Apparently, the model with a constant, and thus higher,
$S_{u,max}$ stores water in the root zone, reducing recharge to the groundwater reservoir that
maintains the lower flows and buffering more water, reducing the peaks. This can also be
clearly seen from the hydrographs (Figure 10), where the later part of the recession in the late-
summer months is much better captured by the time-dynamic model. Nevertheless, the peaks
are too high for the time-dynamic model, which here is linked to an insufficient representation
of snow-related processes, as can be seen from the hydrograph  (April-May) as well, and
possibly by an inadequate interception growth function, both leading to too high amounts of
effective precipitation entering the root zone. An adjustment of these processes would have
resulted in less infiltration and a smaller root zone storage capacity.
The probabilities of improvement for the signatures in Hubbard Brook WS5 show an even
less clear signal, the model cannot clearly identify a preference for either a dynamic or static
formulation of $S_{u,max}$. This absence of a clear preference can be related to the observed
patterns in water balance derived $S_R$ (Figure 4c), which does not show a very clear signal after
deforestation as well, indicating that the root zone storage capacity is of less importance in
this humid region characterized by limited seasonality. Nevertheless, a similar argument  as





for the Hubbard Brook WS2 can be made here, as can be noted that the low flow statistics
(e.g. $Q_{95}$, LFR) slightly improve, and some statistics concerning peak flows deteriorate (e.g.
Peaks, AC), indicating similar issues regarding the modelling of snow and interception.
**5  Conclusion**
In this study, three deforested catchments (HJ Andrews WS1, Hubbard Brook WS2 and WS5)
were investigated to assess the dynamic character of root zone storage capacities using water
balance, trend analysis, four different hydrological models and one modified model version.
Root zone storage capacities were estimated based on a simple water balance approach.
Results demonstrate a good correspondence between water-balance derived root zone storage
capacities and values obtained by a 2-year moving window calibration of four distinct
hydrological models
There are significant changes in root zone storage capacity after deforestation, which were
detected by both, a water-balance based method and the calibration of hydrological models.
We found a good correspondence between water-balance derived root zone storage capacities
and values obtained by a 2-year moving window calibration of four distinct hydrological
models. More specifically, root zone storage capacities showed a sharp decrease in root zone
storage capacities immediately after deforestation with a gradual recovery towards a new
equilibrium. This could to a large extent explain post-treatment changes to the hydrological
regime. Trend analysis suggested recovery times between 5-13 years for the three catchments
under consideration.
These findings underline the fact that root zone storage capacities in hydrological models,
which are more often than not treated as constant in time, may need time-dynamic
formulations with reductions after logging and gradual regrowth afterwards. Therefore, one of
the models was subsequently formulated with a time-dynamic description of root zone storage
capacity. Particularly under climatic conditions with pronounced seasonality and phase shifts
between precipitation and evaporation, this resulted in improvements in model performance
as evaluated by 28 hydrological signatures.
Even though this more complex system behavior may lead to extra unknown growth
parameters, it has been shown here that a simple equation, reflecting the long-term growth of
the system, can already suffice for a time-dynamic estimation of this crucial hydrological



parameter. Therefore, this study clearly shows that observed changes in runoff characteristics
after land use changes can be linked to relatively simple time-dynamic formulations of
vegetation related model parameters.
**Acknowledgements**
We would like to acknowledge the European Commission FP7 funded research project
"Sharing Water-related Information to Tackle Changes in the Hydrosphere– for Operational
Needs" (SWITCH-ON, grant agreement number 603587), as this study was conducted within
the context of SWITCH-ON as an example of scientific potential when using open data for
collaborative research in hydrology.
Open Data were provided by the the Hubbard Brook Ecosystem Study (HBES), which is a
collaborative effort at the Hubbard Brook Experimental Forest which is operated and
maintained by the USDA Forest Service Northern Research Station, Newtown Square, PA,
USA.
Open Data were provided by the HJ Andrews Experimental Forest research program, funded
by the National Science Foundation's Long-Term Ecological Research Program (DEB 08-
23380), US Forest Service Pacific Northwest Research Station, and Oregon State University.





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



1    Table 1. Overview of the catchments and their sub-catchments (WS).

| Catchment | Deforestation period | Treatment | Area [km$^2$] | Affected Area [%] | Aridity index [-] | Prec. [mm/year] | Discharge [mm/year] | Pot. [mm/year] |
|---|---|---|---|---|---|---|---|---|
| HJ Andrews WS1 | 1962 -1966. | Burned 1966 | 0.956 | 100 | 0.39 | 2305 | 1361 | 902 |
| HJ Andrews WS2 | - | - | 0.603 | - | 0.39 | 2305 | 1251 | 902 |
| Hubbard Brook WS2 | 1965-1968 | Herbicides | 0.156 | 100 | 0.57 | 1471 | 1059 | 784 |
| Hubbard Brook WS3 | - | - | 0.424 | - | 0.54 | 1464 | 951 | 787 |
| Hubbard Brook WS5 | 1983-1984 | No treatment | 0.219 | 87% | 0.51 | 1518 | 993 | 746 |

3    Table 2. Applied parameter ranges for root zone storage derivation

| Catchment | $I_{max,eq}$ [mm] | $I_{max,change}$ [mm] | $T_r$ [days] |
|---|---|---|---|
| HJ Andrews WS1 | 1-5 | 0-5 | 0-3650 |
| HJ Andrews WS2 | 1-5 | - | - |
| Hubbard Brook WS2 | 1-5 | 5-10 | 0-3650 |
| Hubbard Brook WS3 | 1-5 | - | - |
| Hubbard Brook WS5 | 1-5 | 0-5 | 0-3650 |





Table 3. Overview of the hydrological signatures

| Signature | Description | Reference |
|---|---|---|
| $Q_{MA}$ | Mean annual runoff | |
| AC | One day autocorrelation coefficient | Montanari and Toth (2007) |
| $AC_{summer}$ | One day autocorrelation the summer period | Euser et al. (2013) |
| $AC_{winter}$ | One day autocorrelation the winter period | Euser et al. (2013) |
| RLD | Rising limb density | Shamir et al. (2005) |
| DLD | Declining limb density | Shamir et al. (2005) |
| $Q_5$ | Flow exceeded in 5% of the time | Jothityangkoon et al. (2001) |
| $Q_{50}$ | Flow exceeded in 50% of the time | Jothityangkoon et al. (2001) |
| $Q_{95}$ | Flow exceeded in 95% of the time | Jothityangkoon et al. (2001) |
| $Q_{5,summer}$ | Flow exceeded in 5% of the summer time | Yilmaz et al. (2008) |
| $Q_{50,summer}$ | Flow exceeded in 50% of the summer time | Yilmaz et al. (2008) |
| $Q_{95,summer}$ | Flow exceeded in 95% of the summer time | Yilmaz et al. (2008) |
| $Q_{5,winter}$ | Flow exceeded in 5% of the winter time | Yilmaz et al. (2008) |
| $Q_{50,winter}$ | Flow exceeded in 50% of the winter time | Yilmaz et al. (2008) |
| $Q_{95,winter}$ | Flow exceeded in 95% of the winter time | Yilmaz et al. (2008) |
| Peaks | Peak distribution | Euser et al. (2013) |
| $Peaks_{summer}$ | Peak distribution summer period | Euser et al. (2013) |
| $Peaks_{winter}$ | Peak distribution winter period | Euser et al. (2013) |
| $Q_{peak,10}$ | Flow exceeded in 10% of the peaks | |
| $Q_{peak,50}$ | Flow exceeded in 50% of the peaks | |
| $Q_{summer,peak,10}$ | Flow exceeded in 10% of the summer peaks | |
| $Q_{summer,peak,50}$ | Flow exceeded in 10% of the summer peaks | |
| $Q_{winter,peak,10}$ | Flow exceeded in 10% of the winter peaks | |



| | | |
|---|---|---|
| $Q_{winter,peak,50}$ | Flow exceeded in 50% of the winter peaks | |
| SFDC | Slope flow duration curve | Yadav et al. (2007) |
| LFR | Low flow ratio ($Q_{90}/Q_{50}$) | |
| FDC | Flow duration curve | Westerberg et al. (2011) |
| $AC_{serie}$ | Autocorrelation series (200 days lag time) | Montanari and Toth (2007) |





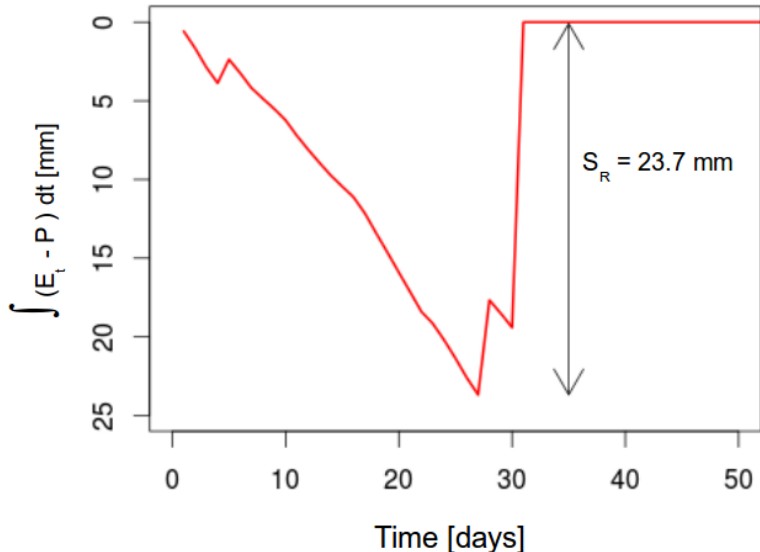

Figure 1. Derivation of root zone storage capacity ($S_r$) for one specific time period in the
Hubbard Brook WS2 catchment as difference between the cumulative transpiration ($E_t$) and
the cumulative effective precipitation ($P_E$).



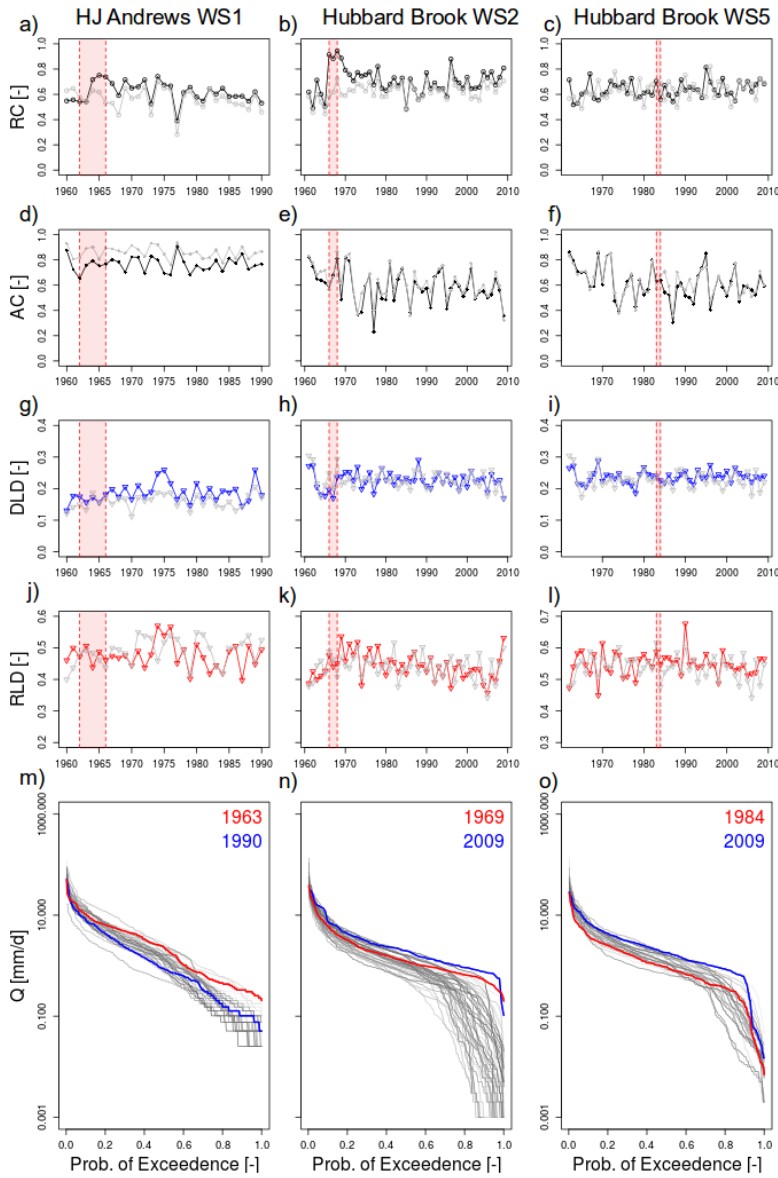

Figure 2. Evolution of signatures in time of a-c) the runoff coefficient, d-f) the 1-day
autocorrelation, g-i) the declining limb density, j-l) the rising limb density with the reference
watersheds in grey and periods of deforestation in red shading. The flow duration curves for
HJ Andrews WS1, Hubbard Brook WS2 and Hubbard Brook WS5 are shown in m-o), where
years between the first and last year are colored from lightgray till darkgray progressively in
time.




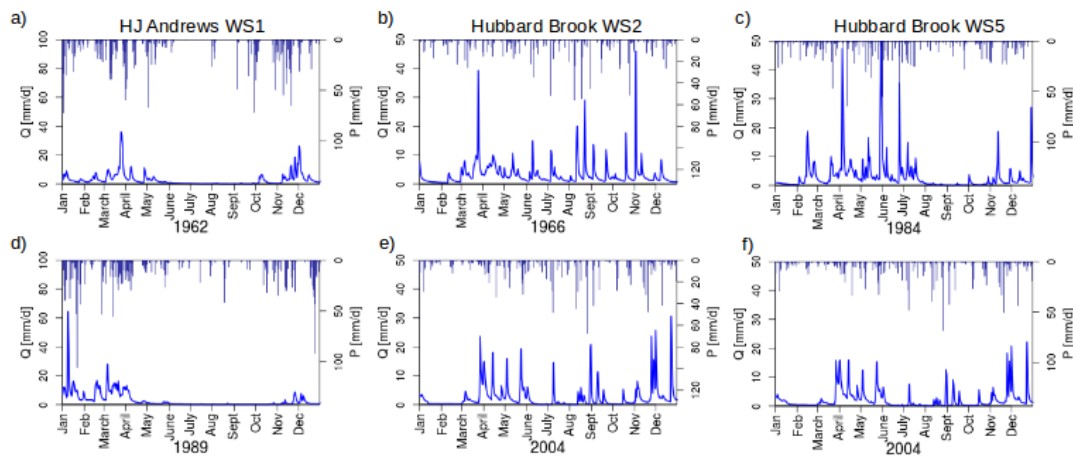

Figure 3. Hydrographs for HJ Andrews WS1 in a) 1963 (annual precipitation $P_A$=2018 mm yr$^{-1}$
, $E_{p,A}$= 951 mm yr$^{-1}$ ) and b) 1989 ($P_A$= 1752 mm yr$^{-1}$, $E_{p,A}$= 846 mm yr$^{-1}$ ), Hubbard Brook
WS2 in c) 1966 ($P_A$ = 1222 mm yr$^{-1}$, $E_{p,A}$ = 788 mm yr$^{-1}$ and d) 2004 ($P_A$ = 1296 mm yr$^{-1}$,
annual $E_{p,A}$ = 761 mm yr$^{-1}$ and Hubbard Brook WS5 in e) 1984 ($P_A$=1480 mm yr$^{-1}$, annual
$E_{p,A}$ = 721 mm yr$^{-1}$ ) and f) 2004 ($P_A$= 1311 mm yr$^{-1}$, $E_{p,A}$ = 731 mm yr$^{-1}$).




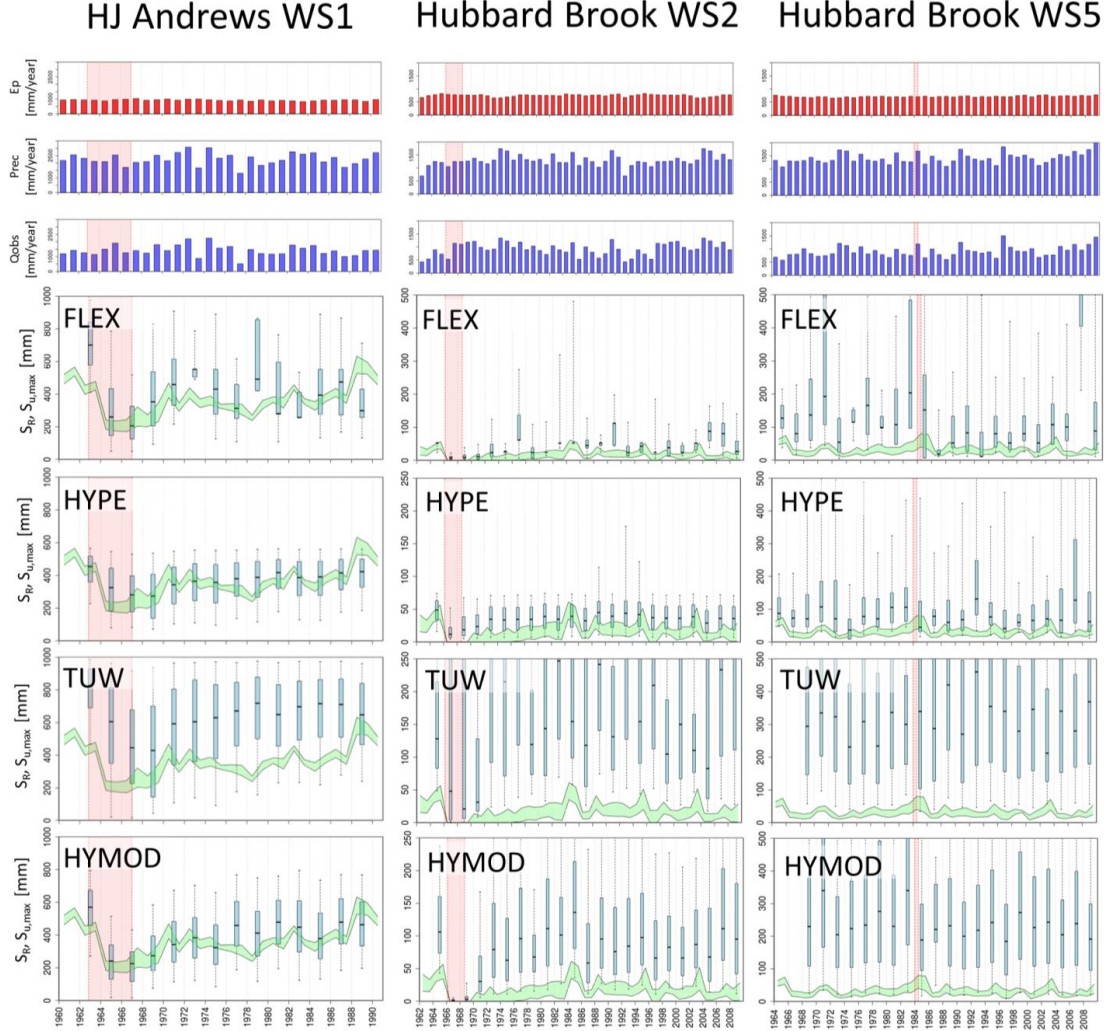

Figure 4. Evolution of root zone storage capacity $S_{R,1yr}$ from water balance-based estimation
(green shaded area, a range of solutions due to the sampling of the unknown interception
capacity) compared with $S_{u,max,2yr}$ estimates obtained from the calibration of four models
(FLEX, HYPE, TUW, HYMOD; blue boxplots) for a) HJ Andrews WS1, b) Hubbard Brook
WS2 and c) Hubbard Brook WS5. Red shaded areas are periods of deforestation.



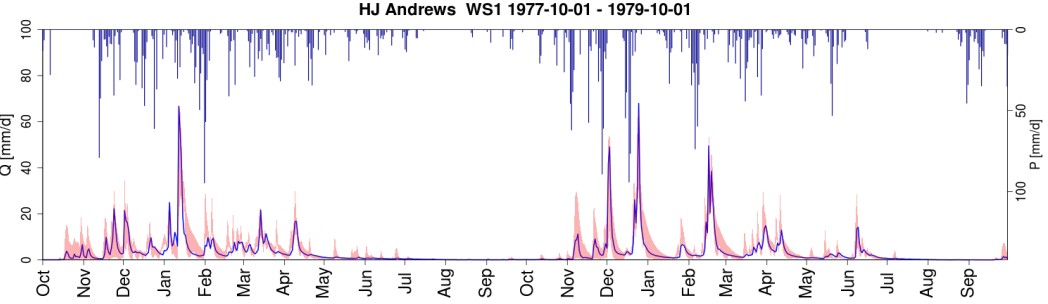

Figure 5. Observed and modelled hydrograph for HJ Andrews WS1 the years of 1978 and
1979, with the red colored area indicating the 5/95% uncertainty intervals of the modelled
discharge. Blue bars show daily precipitation.

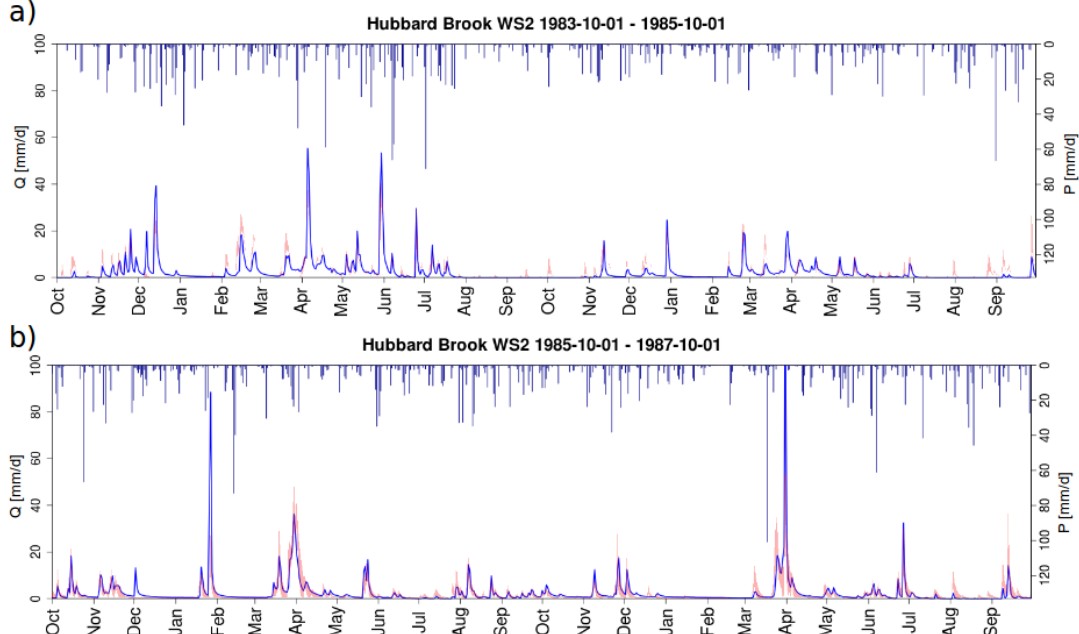

Figure 6. Observed and modelled hydrograph for Hubbard Brook WS2 for a) the years of
1984 and 1985 and b) the years of 1986 and 1987, with the red colored area indicating the
5/95% uncertainty intervals of the modelled discharge. Blue bars show daily precipitation.



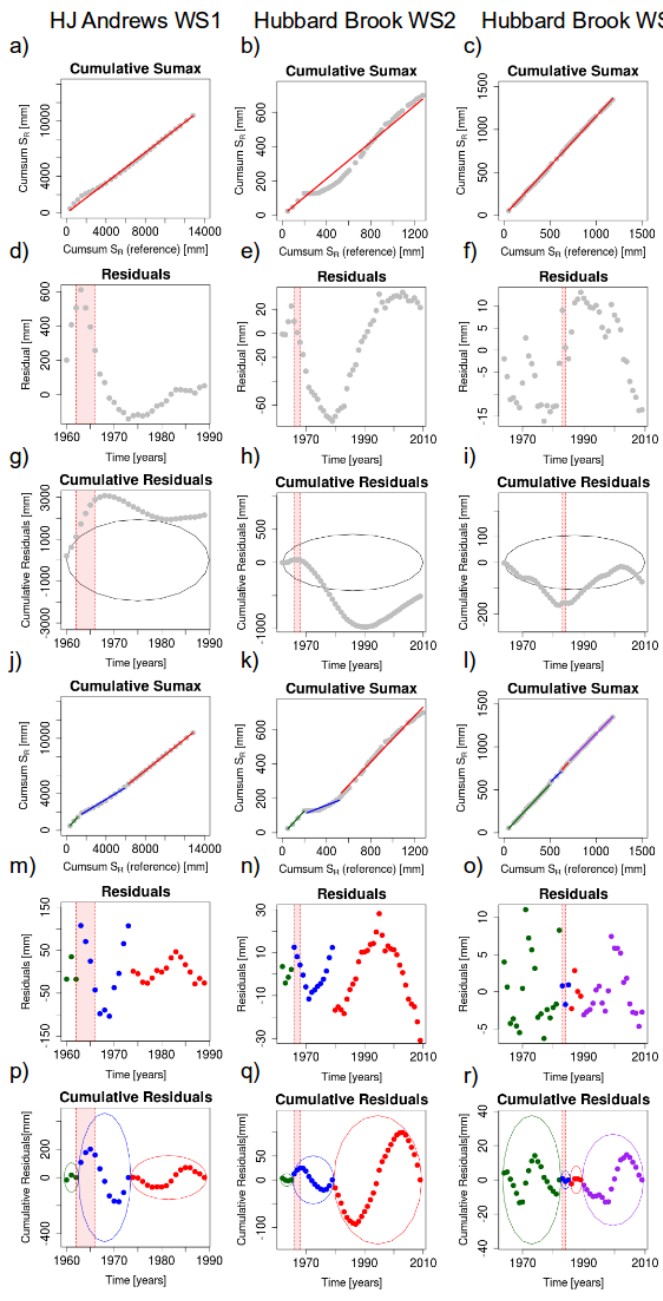

Figure 7. Trend analysis for $S_{R,1yr}$ in HJ Andrews WS1, Hubbard Brook WS2 and WS5 based
on comparison with the control watersheds with a-c) Cumulative root zone storages ($S_{R,1yr}$)



with regression, d-f) residuals of the regression of cumulative root zone storages, g-i)
significance test; the cumulative residuals do not plot within the 95%-confidence ellipse,
rejecting the null-hypothesis that the two time series are homogeneous, j-l) piecewise linear
regression based on break points in residuals plot, m-o) residuals of piecewise linear
regression, p-r) significance test based on piecewise linear regression with homogeneous time
series of $S_{R,1yr}$. The different colors (green, blue, red, violet) indicate individual homogeneous
time periods.



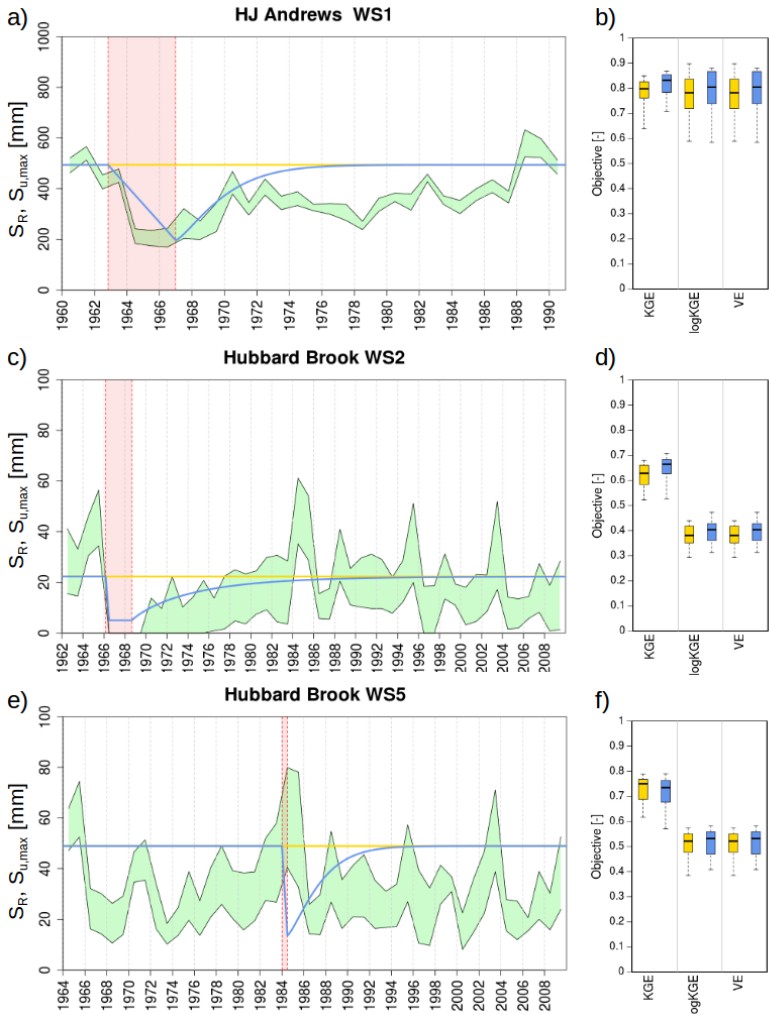

Figure 8. The time invariant $S_{u,max}$ formulation represented by $S_{R, 20yr}$ (yellow) and time
dynamic $S_{u,max}$ fitted Weibull growth function (blue) with a linear reduction during
deforestation (red shaded area) and mean 20-year return period root zone storage capacity $S_{R,}$
$_{20yr}$ as equilibrium value for a) HJ Andrews WS1 with $a=0.0001\ days^{-1}$, $b=1.3$ and $S_{R, 20yr}$ =
$494\ mm$ with b) the objective function values, c) Hubbard Brook WS2 with $a=0.001\ days^{-1}$,
$b=0.9$ and $S_{R, 20yr}$ = $22\ mm$ with d) the objective function values, and e) Hubbard Brook WS5
with $a=0.001\ days^{-1}$, $b=0.9$ and $S_{R, 20yr}$ = $49\ mm$ and with f) the objective function values.
The green shaded area represents the maximum and minimum boundaries of $S_{R,1yr}$ from the
water balance-based estimation, caused by the sampling of interception capacities.


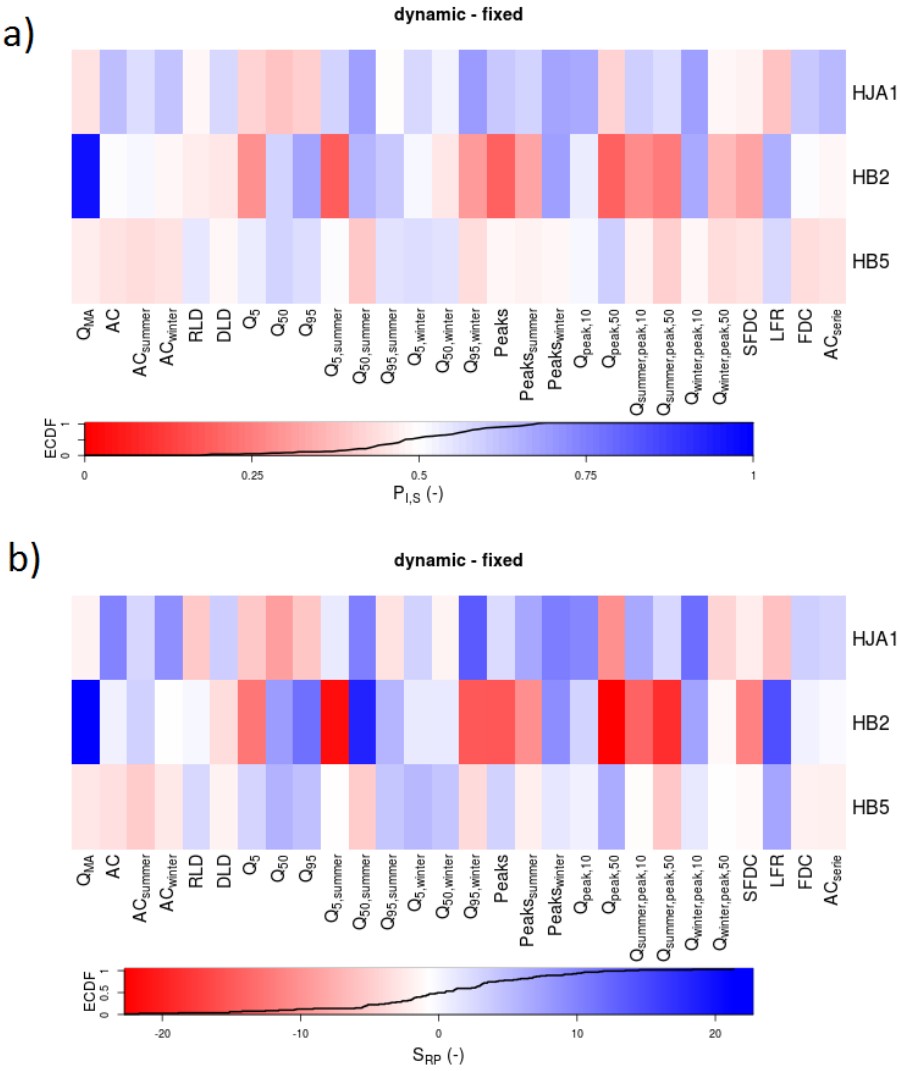

Figure 9. Signature comparison between a time-dynamic and time-invariant formulation of
root zone storage capacity in the FLEX model with a) probabilities of improvement and b)
Ranked Probability Score for 28 hydrological signatures for HJ Andrews WS1 (HJA1),
Hubbard Brook WS2 (HB2) and Hubbard Brook WS5 (HB5). High values are shown in blue,
whereas a low values are shown in red.



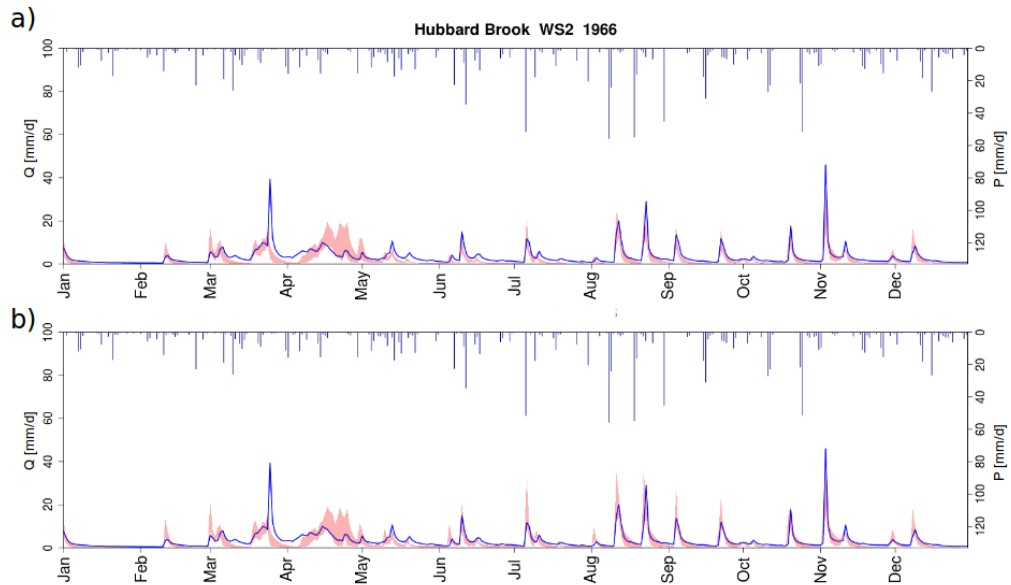

Figure 10. Hydrograph of Hubbard Brook WS2 with the  observed discharge (blue) and the
modelled discharge represented by the 5/ 95% uncertainty intervals (red), obtained with a) a
constant  representation  of  the  root  zone  storage  capacity  $S_{u,max}$  and  b)  a  time-varying
representation of the root zone storage capacity $S_{u,max}$. Blue bars indicate precipitation.

