# Peer review of "The evolution of root zone moisture capacities after"

_Hydrology and Earth System Sciences, 2016_

## Referee Comment (RC1) · Anonymous Referee #1 · 11 Sep 2016

The study of Nijzink et al. investigates the role of root zone moisture capacity on hydrological functioning, in particular after land use change (deforestation). The set-up is clear and the manuscript is well-written. I think this paper fits well in HESS, although some clarifications are needed.

General;

1) The first thing that struck me when getting introduced to the catchments that were used in this study (Table 1) is that the water balances are not closing. For the Hubbard catchments this is hard to check since only PET is given and AET will be lower, for the HJ Andrews catchments, on the other hand, water is 'lost'. Of course it is not a big surprise that a water balance is not closing, given the uncertainty in the observations,
but it becomes tricky when the water balance is used to determine the moisture storage capacity (although you could say that this is also the case for hydrological models that are based on the water balance and that are calibrated on such data). The potential 'disinformation' in observations might influence your estimation of Su,max. I would at least expect a discussion of this potential source of uncertainty, and an estimate of the influence on the results.

2) Lines 7-18 on page 10 show a difficulty of the water-balance method to identify Su,max; you have to assume no storage change. The Introduction describes the importance of flexible Su,max for changing conditions; e.g. land-use change and climate change. And this is where it becomes difficult; under a changing climate (no steady state conditions) you can no longer assume that there is no storage change. In other words; to me it seems that the method to identify Su,max based on the water balance is not applicable in a changing climate.

3) As a proof of concept, a model was included with a dynamic Su,max, which was calibrated by expert-eye to fit the SR1yr-values that were obtained by the water balance method. I agree that a proof of concept is a first step in increasing the process-representation in hydrological models. I would, however, appreciate it if the authors would provide the reader with some suggestions on how to incorporate a dynamic Su,max 'more correctly' in hydrological models. Generally, I am in favor in improving realism in hydrological models, but, extra parameters imply extra uncertainty and the uncertainty should not overwhelm the (hopefully) improved model efficiency. The water balance method seems not feasible in non-steady-state conditions. Do the authors have any suggestions on how to include a dynamic Su,max, or suggestions on observations that could help in this respect?

4) Based on the remarks above, I would suggest to add a separate section to place the results in context (a sort of Discussion, but then different from the one that is included now in the Results section).

[Figure]

Detailed;

a) I know that in the work op Gao and de Boer-Eusink it is shown that climate mainly dictates Su,max rather than the soil. It is, however, maybe valuable to have a look at some of the work of Ilja van Meerveld, who investigated the effect of land use change on soil properties, where it is discussed that the hydraulic conductivity changes as a result of land use change. Could it be possible that the changes in Su,max that you find could actually be assigned to the wrong assumption that the Ksat does not change after land-use change? There are, of course, more parameters in a hydrological model besides a constant moisture storage capacity, that might actually not be completely constant. How can you be sure that the effect you find can only be assigned to the root zone storage and not other parameters?

b) In the calibration of the four hydrological models, two Kling-Gupta terms and the Volumetric Efficiency are used as objective function. As far as I can see, the volume error is already included in the KGE by means of the bias (Beta-term), which would mean that in your calibration strategy, you put extra emphasize on the volume error by explicitly including this term twice (or actually, three times since you use KGE twice). Why is that justified?

c) In your dynamic model, you included extra parameters to describe Su,max, and concluded that it improved the model performance for several indicators. How can you make sure that this improvement is caused by including this process in the model? I would say that for many models you can obtain a (marginal) improvement in model performance by including an extra degree of freedom (an extra parameter), independent of the process that this parameter describes or the realism of the parameterization.

d) I think the research questions in the summary do not exactly reflect the research question in the manuscript (Line 1-5 on page 6).

Overall, some clarifications are needed and some discussion could be added (points mentioned above), but I think that this paper would be of interest for the readers of

HESS.

Kind regards, Lieke Melsen (Wageningen University)

---

## Referee Comment (RC2) · Anonymous Referee #2 · 16 Sep 2016

The authors introduced a method to estimate the change of the root zone storage capacities after land use change in three experimental catchments. The topic of this paper is in the scope of HESS, and relevant. Overall the paper is well-written. Some corrections and suggestions for improvement are listed below.

**General comments**

- In general, I find the paper too long. Maybe some details of the methodology can be moved into the Supplementary Material.
- I suggest to be more precise in the title. First, ending the title by "under change" seems quite strange to me. Is it still land use change, or climate change or other ? (same remark at line 10 of page 2). Then, "predictions" is too vague because it can be applied to many processes (prediction of discharge, of flood, of vegetation dynamics...). In addition, more discussion on the potential applications with this kind of method is needed in the conclusion and perspectives.
- The results and the figures, which include many hydrological signatures, are not always simple to read and to analyze. Then, the interest of the discussion can be lost during the reading of Section 4. Thus, I would recommend to split this section in 2 sections to distinguish Results and Discussion.

**Specific comments**

**Abstract**
1/ "long-term data" => you can be more precise
2/ line 24 of page 2: "better representations of high flows and peak flows" => what about the low flows ?

**Introduction**
3/ To be more precise, the vegetation partitions first precipitation into interception, stemflow and throughfall. Then, the fraction of rainfall that reaches the surface is partitioned into evapotranspiration, drainage and also surface runoff.
4/ line 28 of page 3: the year is missing for Vose et al. and also in the References section.
5/ line 10 of page 4: interception/soil evaporation/transpiration and surface runoff/drainage
6/ line 21 of page 4: "system" is unclear. Please reformulate.
7/ lines 30-32 of page 4: The sentence is difficult to read. Please rewrite.
8/ lines 6-7 of page 5: $S_R$ has already been defined in page 3, line 15. The best is to combine "sometimes also referred to as plant available water holding capacity" with the text in line 15 of page 3.
9/ lines 18-21 of page 5: the sentences are very unclear. Please reformulate.
10/ lines 3-4 of page 6: words are missing in the 2$^{nd}$ hypothesis formulation, please check.

**Section 2**
11/ In each sub-sections, the references to Table 1 for watershed characteristics should be merged and written once in the section, just before sub-section 2.1. Then, the references at lines 12, 19-20 of page 6 and lines 1-2 of page 7 can be removed.

**Section 3**
12/ lines 14-17 of page 9: For long-term mean variables: $E_t$ => $\overline{E_t}$. The same for Q and $E_p$.
13/ line 5 of page 10: "obtained by equation 6" => "obtained by equation 7"
14/ lines 7-9 of page 10: this is a strong assumption, especially under climate change where the water storage changes. This point should be more discussed when the method based on the water balance is applied.

15/ line 11 of page 11: "FLEX-based model" => "The FLEX-based model"

16/ line 1 of page 12: this process is not represented in Figure S2.

17/ line 9 of page 12: what are the fluxes ? Moreover, transpiration is indicated in the text but "Evaporation" is written in Figure S3. Please, check the coherency between the text and the Figure.

18/ line 11 of page 13: what is n ?

19/ line 4 page 14: $Z_{95}$ should be $Z_{p95}$

20/ line 2 page 16: "Table 2" => "Table 3"

**Section 4**

21/ lines 23-24 of page 17: this is not particularly obvious in Figure 2f.

22/ lines 20-21 of page 24: I do not see this improvement on Figure 10, maybe due to the scale of the plots.

**Table/Figures**

23/ Table 1:
- I would add a column for the abbreviations of each catchment, as used in figure 9 (see my comment hereafter for the whole text).
- "Precip" should be "Precipitation".
- what is "Pot." ? It is the potential evaporation?
- remove "%" from 87% in the last line.

24/ Table 3: the reference for Jothityangkoon et al. (2001) is missing in the References section.

25/ Figure 1: in the label of y-axis, "P" should be "$P_E$"

**Supplementary material**

26/ Table S1: please check the Imax values (Min=Max=0 !)

27/ Figure S2:
- replace "Snow" term in the figure by "S".
- Peff and interception are not represented in the Figure.
- q3 should be replaced by q2 in the figure.

28/ Table S2: the wilting point cannot be higher than the field capacity. Please check the max values.

29/ Figure S3:
- replace "Snow" term in the figure by "S".
- q3 should be replace by q2 in the figure.
- Q should be replace by Qf.
- what is dq ?

30/ Figure S4: the surface runoff is missing.

**In the whole text**
- choose between "parameterization" and "parametrization"
- I suggest to use the abbreviations of the catchments in the text, as used in figure 9. It will facilitate the reading of the paper.
- there is a confusion all along the text when the term "evaporation" is used. The term "Evapotranspiration", which is the sum of soil evaporation, interception evaporation and transpiration, is more adequate.

---

## Author Comment (AC1) · 17 Sep 2016

We would like to thank Lieke Melsen for her constructive comments. We will try to improve on the raised issues.

"The first thing that struck me when getting introduced to the catchments that were used in this study (Table 1) is that the water balances are not closing. For the Hubbard catchments this is hard to check since only PET is given and AET will be lower, for the HJ Andrews catchments, on the other hand, water is 'lost'. Of course it is not a big surprise that a water balance is not closing, given the uncertainty in the observations, but it becomes tricky when the water balance is used to determine the moisture storage capacity (although you could say that this is also the case for hydrological models that

are based on the water balance and that are calibrated on such data). The potential 'disinformation' in observations might influence your estimation of Su,max. I would at least expect a discussion of this potential source of uncertainty, and an estimate of the influence on the results."

This is a very valid point. We relate the fact that the water balance does not close mainly to the calculation of the potential evaporation, which here, due to data availability, was estimated from temperature only. We will add a paragraph in the discussion on the consequences of these uncertainties for the estimation of SR.

"Lines 7-18 on page 10 show a difficulty of the water-balance method to identify Su,max; you have to assume no storage change. The Introduction describes the importance of flexible Su,max for changing conditions; e.g. land-use change and climate change. And this is where it becomes difficult; under a changing climate (no steady state conditions) you can no longer assume that there is no storage change. In other words; to me it seems that the method to identify Su,max based on the water balance is not applicable in a changing climate."

We agree with the statement that under changing conditions storage may change. Nevertheless, in the applied method the water balance is merely used to derive an estimate of average transpiration rates. Therefore, we argue that under changing conditions, this estimate is an upper limit of the actual transpiration, whereas in reality it may be lower. In addition, a long-term water balance would not reflect the yearly variations in climate, whereas rather short term water balances may be influenced by storage changes. This is also why, in a trade-off and to keep the effects of storage change as low as possible, the water balances over 2-year periods were used. To substantiate this, to put into context and to assess the effect of storage change, please see Figure 1 below, where, for comparative reasons, we additionally estimated Su,max using a 5-year window to further reduce the influence of storage changes. It can be noted here that the green shaded area, representing the water balance-based estimates, is flatter compared to the results obtained with the 2-year water balance (maximum 500mm

compared to 600mm in Figure 4 of the manuscript). This is due to more averaging by taking a longer period for the water balance estimation. In spite of that, the general patterns hold, and in our opinion supports our results. Eventually, we would like to point at the results obtained in the undisturbed reference (or control) watersheds, in Figure S8 of the Supplementary Material. These results are obtained in absence of any land use change, and thus reflect only the changes due to climatic variability (and are thus a proxy for climate influenced inter-annual storage changes). The different pattern compared to the deforested catchments then indicates the isolated effects of storage change due to deforestation and thus transpiration (under the assumption that both control and deforested catchments were subject to the same climate variability). Thus, we would argue that the changes in storage that may occur, are relatively small compared to the annual fluxes of precipitation and discharge.

"As a proof of concept, a model was included with a dynamic $S_{u,max}$, which was calibrated by expert-eye to fit the SR1yr-values that were obtained by the water balance method. I agree that a proof of concept is a first step in increasing the process representation in hydrological models. I would, however, appreciate it if the authors would provide the reader with some suggestions on how to incorporate a dynamic $S_{u,max}$ 'more correctly' in hydrological models. Generally, I am in favor in improving realism in hydrological models, but, extra parameters imply extra uncertainty and the uncertainty should not overwhelm the (hopefully) improved model efficiency. The water balance method seems not feasible in non-steady-state conditions. Do the authors have any suggestions on how to include a dynamic $S_{u,max}$, or suggestions on observations that could help in this respect?"

We would like to suggest simple conceptual formulations of growth dynamics, similar to the growth function applied in this case. This would lead to the addition of, at most, three new parameters. These could be free calibration parameters, but we agree that this may lead to additional uncertainty. And even though the water balance method may only give an estimation of the dynamics of the root zone storage capacity, this

method may prove valuable to derive at least some information about the \*shape\* of the growth curve. It can also be noted that transpiration estimates are derived from the water balance in this case, but there are also (remote-sensed) products available to estimate the transpiration. In this way, issues with water balances that may not close are fully avoided.

"Based on the remarks above, I would suggest to add a separate section to place the results in context (a sort of Discussion, but then different from the one that is included now in the Results section)."

We will add a separate section in the discussion about the uncertainties that are introduced by 1) data used in the water balance, 2) storage changes affecting the water balance. In addition, we will elaborate in Section 4.4 on how to explicitly apply our findings in conceptual modelling.

"I know that in the work op Gao and de Boer-Eusink it is shown that climate mainly dictates Su,max rather than the soil. It is, however, maybe valuable to have a look at some of the work of Ilja van Meerveld, who investigated the effect of land use change on soil properties, where it is discussed that the hydraulic conductivity changes as a result of land use change. Could it be possible that the changes in Su,max that you find could actually be assigned to the wrong assumption that the Ksat does not change after land-use change? There are, of course, more parameters in a hydrological model besides a constant moisture storage capacity, that might actually not be completely constant. How can you be sure that the effect you find can only be assigned to the root zone storage and not other parameters?"

Indeed, there is no absolute certainty that other parameters are not affected by the land use change. Nevertheless, when vegetation is removed, it is not inconceivable to assume that the vegetation-related parameters are considerably affected. This can also be seen from the posterior-distributions of the other parameters, see the Supplementary Material. In the 2-year window calibration, all parameters were left for calibration,

and they all had the freedom to change over time. Nevertheless, the root zone storage capacity showed the most dynamical character, whereas others remained more constant in time. In addition, we would expect that changes in hydraulic conductivity are tightly linked to changes in porosity. In other words, an increase of porosity is not unlikely to decrease the flow resistances and thus increase Ksat, while simultaneously reducing the storage capacity. It must also be noted that hydraulic conductivity Ksat cannot be compared directly to any of the catchment scale conceptual model parameters applied here.

"In the calibration of the four hydrological models, two Kling-Gupta terms and the Volumetric Efficiency are used as objective function. As far as I can see, the volume error is already included in the KGE by means of the bias (Beta-term), which would mean that in your calibration strategy, you put extra emphasize on the volume error by explicitly including this term twice (or actually, three times since you use KGE twice). Why is that justified?"

This is a valid point; we will compare the outcomes with a calibration based on a combination of KGE and logKGE to test how much this influences our results.

"In your dynamic model, you included extra parameters to describe Su,max, and concluded that it improved the model performance for several indicators. How can you make sure that this improvement is caused by including this process in the model? I would say that for many models you can obtain a (marginal) improvement in model performance by including an extra degree of freedom (an extra parameter), independent of the process that this parameter describes or the realism of the parameterization."

To avoid this, both model approaches were given the same number of degrees of freedom. In other words, both models had the same number of free calibration parameters. This is why the growth functions were fixed, and not left for calibration.

"I think the research questions in the summary do not exactly reflect the research question in the manuscript (Line 1-5 on page 6)."

[Figure]

We will rephrase it to be more consistent throughout the manuscript.
[Figure]

**HJ Andrews WS1**

Figure area with plot showing Sumax [mm] on y-axis (0 to 1000) and years 1958–1988 on x-axis.

**Fig. 1.** Evolution of root zone storage capacity SR,1yr from a 5-year water balance-based estimation (green shaded area) compared with estimates obtained from the calibration of FLEX (blue boxplots).

---

## Author Comment (AC2) · 1 Oct 2016

We would like to thank Anonymous Referee 2 for his/her feedback. We will try to improve on the raised issues.

**General comments**

*"In general, I find the paper too long. Maybe some details of the methodology can be moved into the Supplementary Material"*

Agreed. We will shorten some parts of the manuscript.

*"I suggest to be more precise in the title. First, ending the title by "under change""*

*seems quite strange to me. Is it still land use change, or climate change or other ? (same remark at line 10 of page 2). Then, "predictions" is too vague because it can be applied to many processes (prediction of discharge, of flood, of vegetation dynamics...). In addition, more discussion on the potential applications with this kind of method is needed in the conclusion and perspectives."*

We rephrased the title to: "The evolution of root zone moisture capacities after deforestation: a step towards hydrological predictions under land use change?". In addition, we will add a discussion on practical applications of the method in conceptual modelling (also suggested by Referee 1).

*"The results and the figures, which include many hydrological signatures, are not always simple to read and to analyze. Then, the interest of the discussion can be lost during the reading of Section 4. Thus, I would recommend to split this section in 2 sections to distinguish Results and Discussion."*

We decided to merge the results and discussion in order to avoid repetition and to make the article more concise. We still prefer to keep it like this, also with regard to the first comment (the paper is still rather long). Nevertheless, we will have a critical look at the figures and discussion, and will try to clarify wherever we can.

**Specific comments**

**Abstract**

*1/ "long-term data" => you can be more precise*

*2/ line 24 of page 2: "better representations of high flows and peak flows" => what about the low flows ?*

1. We changed it to "long-term data (30-40 years of observations)"

2. The low flows improved for the Hubbard Brook catchments, whereas the low flows

did not show improvements in the HJ Andrews catchment. See also page 24, lines 13-26.

**Introduction**

*3/ To be more precise, the vegetation partitions first precipitation into interception, stemflow and throughfall. Then, the fraction of rainfall that reaches the surface is partitioned into evapotranspiration, drainage and also surface runoff.*

*4/ line 28 of page 3: the year is missing for Vose et al. and also in the References section.*

*5/ line 10 of page 4: interception/soil evaporation/transpiration and surface runoff/drainage*

*6/ line 21 of page 4: "system" is unclear. Please reformulate.*

*7/ lines 30-32 of page 4: The sentence is difficult to read. Please rewrite.*

*8/ lines 6-7 of page 5: SR has already been defined in page 3, line 15. The best is to combine "sometimes also referred to as plant available water holding capacity" with the text in line 15 of page 3.*

*9/ lines 18-21 of page 5: the sentences are very unclear. Please reformulate.*

*10/ lines 3-4 of page 6: words are missing in the 2nd hypothesis formulation, please check.*

3. We fully agree, and we rephrased the first sentence to be more correct.

4. We corrected this.

5. We rephrased it into "runoff components and evaporation", as we tried to lump the terms together that you refer to.

6. We changed it to "hydrological system"
7. We rephrased this.

8. We changed this and placed the text at page 3, line 15.

9. We rephrased this.

10. We checked and rephrased the sentence.

**Section 2**

*11/ In each sub-sections, the references to Table 1 for watershed characteristics should be merged and written once in the section, just before sub-section 2.1. Then, the references at lines 12, 19-20 of page 6 and lines 1-2 of page 7 can be removed.*

11. We agree with the suggestion and changed this.

**Section 3**

*12/ lines 14-17 of page 9: For long-term mean variables: Et => Et. The same for Q and Ep.*

*13/ line 5 of page 10: "obtained by equation 6" => "obtained by equation 7"*

*14/ lines 7-9 of page 10: this is a strong assumption, especially under climate change where the water storage changes. This point should be more discussed when the method based on the water balance is applied.*

*15/ line 11 of page 11: "FLEX-based model" => "The FLEX-based model"*

*16/ line 1 of page 12: this process is not represented in Figure S2.*

*17/ line 9 of page 12: what are the fluxes ? Moreover, transpiration is indicated in the text but "Evaporation" is written in Figure S3. Please, check the coherency between the text and the Figure.*

*18/ line 11 of page 13: what is n ?*
*19/ line 4 page 14: Z95 should be Zp95*

*20/ line 2 page 16: "Table 2" => "Table 3"*

12. We changed this.

13. We changed this.

14. We agree with this and will add a discussion on this (see also the Response to Referee 1)

15. We changed this.

16. This is not the case for the current set-up. We will remove the sentence.

17. We rephrased it to make it more consistent.

18. n is a weighing exponent. We will clarify this in the text.

19. We changed this.

20. We changed this.

**Section 4**

*21/ lines 23-24 of page 17: this is not particularly obvious in Figure 2f.*

*22/ lines 20-21 of page 24: I do not see this improvement on Figure 10, maybe due to the scale of the plots.*

21. We do agree that the pattern is rather variable over time, but comparing the highest peaks before deforestation with the peaks after deforestation show that the values were higher before deforestation. The same applies to the lower values. Calculating the mean autocorrelation before deforestation and after also confirm this; 0.65 before deforestation and 0.58 after deforestation.

22. More specifically, we are referring to the parts of the hydrograph at the end of June until August. Please note the white space between observation and model in the case of a constant root zone storage capacity, whereas for the dynamic model they overlap.

**Table/Figures**

*23/ Table 1:*

*- I would add a column for the abbreviations of each catchment, as used in figure 9 (see my comment hereafter for the whole text).*

*- "Precip" should be "Precipitation".*

*- what is "Pot." ? It is the potential evaporation?*

*- remove "%" from 87*

*24/ Table 3: the reference for Jothityangkoon et al. (2001) is missing in the References section.*

*25/ Figure 1: in the label of y-axis, "P" should be "PE"*

23. We agree with the suggestions/corrections and will change it. "Pot" refers indeed to potential evaporation.

24. We corrected this.

25. We corrected this.

**Supplementary material**

*26/ Table S1: please check the Imax values (Min=Max=0 !)*

*27/ Figure S2:*

*- replace "Snow" term in the figure by "S".*

*- Peff and interception are not represented in the Figure.*

*- q3 should be replaced by q2 in the figure.*

*28/ Table S2: the wilting point cannot be higher than the field capacity. Please check the max values.*

*29/ Figure S3:*

*- replace "Snow" term in the figure by "S".*

*- q3 should be replace by q2 in the figure.*

*- Q should be replace by Qf.*

*- what is dq ?*

*30/ Figure S4: the surface runoff is missing.*

26. This should be 0 – 5 mm

27. We changed this

28.  These percentages should be added up (they do not represent the actual wilting point and field capacity).  Thus, when wcep is 0.2, and wcfc 0.5, the wilting point is at 0.2 of the soil depth and the field capacity at 0.7 (0.2+0.5).

29. We corrected this and added the missing description of dq.

30. Correct, this model structure does not take overland flow into account.

**In the whole text**

*"–choose between "parameterization" and "parametrization"*

We changed it throughout the whole manuscript to "parameterization"

*"–I suggest to use the abbreviations of the catchments in the text, as used in figure 9.*

*It will facilitate the reading of the paper."*

We will consider this, though this is just a matter of taste. Personally, a text with too many abbreviations may also become harder to read.

*"–there is a confusion all along the text when the term "evaporation" is used. The term "Evapotranspiration", which is the sum of soil evaporation, interception evaporation and transpiration, is more adequate."*

We tried to be consistent throughout the manuscript and refer to evaporation when we mean all the evaporative fluxes. We actually believe that the term "evapotranspiration" should not be used and we would like to refer to Savenije (2004) for arguments to not use this term. Briefly, transpiration is a bio-physical process, with different timescales and characteristics thereby being distinct to all other evaporative fluxes, which are purely physical processes. The term "evapotranspiration" is therefore a misleading definition, adding up different kinds of processes.

**References**

Savenije, H. H. G.: The importance of interception and why we should delete the term evapotranspiration from our vocabulary, Hydrological Processes, 18, 1507-1511, 10.1002/hyp.5563, 2004.

---

## Referee Comment (RC3) · A. Ducharne (Referee) · 14 Oct 2016

**Manuscript hess 2016-427 by Nijzink et al.: "The evolution of root zone moisture capacities after land use change: a step towards predictions under change"**
Review by Agnès Ducharne

The proper way to define the root zone and corresponding storage capacities, whether for water, or carbon, or nutrients, is a very topical research question in environmental science. This question is addressed here from the point of view of water, which is relevant for HESS, and in the framework of land use conversion, to question the need for implementing a dynamical description of the root zone moisture capacity in conceptual hydrological models. The method relies on comparing several such models (including a very simple water balance model) to long-term observations in three experimental catchments having undergone deforestation, and paired with control catchments without land cover change.

This looks like a sound research strategy, but the paper itself suffers from shortcomings, which cast serious doubts, to my opinion, on the relevance of the proposed conclusions. In particular, if the results do show a dependence of the water budget on the land cover (at least in 2/3 catchments), it is not clear from the results if this dependence comes or not from the RZCM. It is driven by the RZCM in the selected models, but this is not systematic (see my comment 2 below), and may be too model-dependent to have a broad meaning (see comment 1a).

**1. We lack a lot of information regarding the models and their use.** The main idea is to propose evolutions of the root zone moisture capacity (RZMC) at a yearly time step by a kind of inverse modelling using the observed river discharge of the perturbed and unperturbed catchments as input.

1.a) The simple "water balance model" allows a direct inversion of the RZMC, given parameters describing the canopy interception processes and the vegetation recovery time, and restricting the water balance to only 5 months between May and October, to get rid from the influence of snow (the experimental catchments are located in Oregon and New Hampshire):
- Unless vegetation growth is really restricted to these 5 months, this tends to underestimate the RZMC, and could explain why the Hubbard Brook estimates are so small for forested sites (23 mm on Figure 1)
- The total evaporation seems to comprise only transpiration and interception loss, and neglect soil evaporation: is it justified?
- Transpiration depends on a potential evaporation, which is not explained in the paper: does potential evaporation depend on the development of the canopy, as could be quantified by the Leaf Area Index (LAI)? This dependence is well known fact, and can be described for instance by the crop coefficient when following the FAO guidelines of Allen et al. (1986), or as a function of LAI like in the SVAT (Soil-Vegetation-Atmosphere Transfers) models. If such dependence exists in the experimental catchments, it should lead transpiration to decrease after deforestation, and recover with vegetation regrowth, with opposite effects on runoff, in agreement with Figure 2(a-c). In this case, if the model overlooks the positive link between vegetation development and the magnitude of transpiration, it should lead to underestimate the decrease of transpiration after deforestation, and to overestimate the decrease of the RZCM to match the increased observed runoff.
- A Monte-Carlo approach is used to assess the effect of the 3 parameters involved in the model (see Table 2) and this allows deriving a very useful uncertainty range around the estimated RZCM. Yet, no justification is given regarding the selected range for these parameters, which is a strong constrain to the uncertainty.

1.b) The other four models are published conceptual hydrological models, and are calibrated over consecutive 2-yr windows to match the observed water discharge. These models seem to describe the full hydrological year, including the periods of snow, which is a significant difference with the previous approach. Even if some information is given in the Supplementary (but not at the same level

for all the models), the reader should find in the main text if the snow is explicitly described, and how the evapotranspiration is calculated (in particular how it depends on the vegetation development, for the same reasons as explained above).

Some details should also be given regarding the calibration itself: How many parameters are calibrated in addition to RZCM (Su,max) for each model? Can all of them change in each 2-yr window, or does only Su,max change? How many tested parameter sets? How many parameter sets are kept at the end of the calibration (equifinality) and what are the corresponding performances to fit the observed discharge? There is a long paragraph from p12L27 to p13L14 which is rather hard to follow for non-specialists of optimization, and could usefully be replaced by objective information regarding the qualities and weakness of the resulting calibration.

1.c) Another model is used, and presented in 3.5. It's an adaptation of FLEX, one of the above four models, in which an a priori rule for RZCM recovery with time after deforestation is added. First, it would probably be clearer if this model was presented just after the others. Second, much information, again, is lacking:
- How is the evolution Imax described since it also varies with time (p15L17-18)?
- How are the parameters a and b of Eq. 11 selected? The resulting values are only given in the caption of Fig8, but don't they deserve some analysis? Do they relate logically to the recovery times that are discussed in section 4.3?
- How is decided when is RZCM minimum, and which is the minimum value, since Eq. 11 only describes the increasing part of the variations shown on Figure 8?
- Fig 8 shows performance criteria with and without the dynamic formulation of Su,max: to which period do they correspond?  We must assume that the period is the full observed period for each catchment, but does it make sense for HB5, where half of the full period is before deforestation? Couldn't it be interesting to test the proposed function over the recovery period only?

**2. The conclusions are too frequently not supported by the Figures.** Examples:
- p17,L3-4: "the three deforested catchments in the two research forests show generally similar response dynamics after the logging of the catchments (Fig.2)." No, for each of the rows/signatures, you can find one outlier over the three catchments.
- p18, L24-26 (regarding Figure 4): "Comparing the water balance and model-derived estimates of root zone storage capacity SR and Su,max, respectively, then showed that they exhibit very similar patterns in the study catchments." This is abusive since TUW and HYMOD completely miss the difference between HJA and HB, and HB5 doesn't show a clear response to deforestation against inter-annual variability for most models. When discussing Figure 4, the focus is put on the differences in RZCM due to deforestation and recovery. Yet, these differences are much smaller than the ones between the sites, and have a similar magnitude as the inter-annual variability for the two Hubbard Brook catchments. This should be taken in consideration in the discussion.
- p20, L23-26: "It can be argued, that a combination of a relatively long period of low rainfall amounts and high potential evaporation, as can be noted by the relatively high mean annual potential evaporation on top of Figure 4b, led to a high demand in1985". But the top three plots on Fig 4 are so small we can't see much!
- p21, L3-4: "Generally, the models applied in Hubbard Brook WS2 show similar behavior as in the HJ Andrews catchment." It's far from being obvious for HB5.
- p22, L16-17: "The results shown in Figure 4 indicate that these catchments had a rather stable root zone storage capacity during deforestation" (for HJA and HB2). Deforestation is indicated by a red band, and we clearly show a decreasing, not stable, RZCM during deforestation in HJA; for HB2, we don't see anything because the y-axis range is too large.
- p23, L24-28: "Evaluating a set of hydrological signatures suggests that the dynamic formulation of Su,max allows the model to have a higher probability to better reproduce most of the signatures tested here (54% of all signatures in the three catchments) as shown in Figure 9a. A similar pattern is obtained for the more quantitative SRP (Figure 9b), where in 52% of the cases improvements are

observed." This is abusive because your get degradation of the performance for 46% of the signatures in Fig9a, and 48% in Fig 9b, which is far from being negligible. If you look at HB5 only, the degraded signatures dominate, which contradicts the conclusion at p24, L27-29.

- p24, L6-7: "In addition, a dynamic formulation of $S_{u,max}$ permits a more plausible representation of the variability in land-atmosphere exchange following land use change". Where does this come from? Provided that no signature in Fig 9 and Table 3 addresses the variability of land-atmosphere exchanges (all the signatures describe elements of the streamflow time series).

- p24, L9-10: "Fulfilling its function as a storage reservoir for plant available water, modelled transpiration is significantly reduced post-deforestation, which in turn results in increased runoff coefficients": if I see well on the very small Fig 2c, the results show exactly the opposite for HB5.

- p24, L19-21: "This can also be clearly seen from the hydrographs (Figure 10), where the later part of the recession in the late summer months is much better captured by the time-dynamic model." Personally, I see exactly the opposite, as the time-varying RZCM model in Fig 10b overestimates the peaks, which is not the case of the constant RZCM model in Fig 10a.

- Finally, the conclusion relies on a selection of the results that support the assumption of the authors, without considering the results that contradict it, and without a hint of doubt. The limits of the approach (including the model dependency, the small sample of observations which are not perfectly consistent) are not all discussed, nor any alternative frameworks. The authors could for instance consider the possibility that the RZCM could remain unchanged but not fully exploited by the vegetation. This is typically what helps some types of vegetation to resist to drought conditions.

**3. Abstract:**
The abstract is not very clear regarding the methods (the proposed method is not solely based on climate data as written at L8-9, but it requires information on the deforestation, based on inverting the discharge observation in the present case). Like the conclusion, it builds too much on overstatement, but there is also an annoying circular reasoning, since the main conclusion comes from the beginning (L5-7: "Often this parameter [RZCM] is considered to remain constant in time. This is not only conceptually problematic, it is also a potential source of error under the influence of land use and climate change.")

**4. Other comments:**
- Trend analysis (method in 3.4, results in 4.3): is it really about trends or about variability? Can we really speak of "trends" on sub-periods as short as those highlighted in blue and green in Fig 7o and 7r? Couldn't these two periods be lumped together? Some references should be given where to find more details on the extraction and interpretation of the 95%-confidence ellipses. Finally, Fig 7 is much too small.
- Some sentences I did not find clear, although the paper is generally well written:
  - p3, L13-15: "By extracting plant available water between field capacity and wilting point, roots create moisture storage volumes within their range of influence."
  - p 4, L7-8: "other species with different water demands may be more in favor in the competition for resources"
  - p4, L15: "These studies found that deforestation often leads to higher seasonal flows". Do you mean higher peak flows?
  - p4, 30-31: "More systematic approaches, thus incorporation the change in the model formulation itself"
  - p14, L28-29: "the calibration was run with a series temporally evolving root zone storage capacities"
  - p26, L27: I suggest using attributed to rather than caused by, unless a clear causality can be demonstrated.

---

## Editor Comment (EC1) · FF Fenicia (Editor) · 26 Oct 2016

Main comments

Dear Authors, the reviewers have provided a generally positive evaluation of your work, and they have identified important areas needing clarification and improvement. I do not think that the reviewers concerns will be too problematic to address.

In addition to the reviewers comments, I would like to point you towards the following paper:

Westra, S., M. Thyer, M. Leonard, D. Kavetski, and M. Lambert (2014), A strategy for diagnosing and interpreting hydrological model nonstationarity, Water Resour. Res.,

50, 5090–5113, doi:10.1002/2013WR014719

Which considers the non-stationarity of the SuMax parameter, and therefore is closely related to your work. I would like to see this paper not only cited but also discussed.

I would prefer to see separate results and discussion sections. Otherwise it is difficult to separate results that are directly supported by your analysis (results section), and interpretations that may go beyond it (discussions).

Other comments

As per HESS guidelines, multi-letter variables should be avoided. In Table 3 for example, you should refer to the signatures with 1 capital and the rest subscripts. E.g. you can call all signatures with the Greek letter gamma, and then subscripts to identify the particular signature.

What is n in Eq 8, 9, 10?

Eq 10: you have $Z_{p95}$ in the equation and $Z_{95}$ in the description. Use consistent notation

Eq 10: $SR,20r$ in the equation and $SR,20yr$ in the description. Also don't use the star as a multiplier. Either the dot or nothing.
* * *

---

## Author Comment (AC3) · 26 Oct 2016

We would like to thank Dr. Ducharne for her feedback on the manuscript. We will try to improve on the comments and raised issues.

*1. We lack a lot of information regarding the models and their use. The main idea is to propose evolutions of the root zone moisture capacity (RZMC) at a yearly time step by a kind of inverse modelling using the observed river discharge of the perturbed and unperturbed catchments as input.*

[Figure]

*1.a) The simple "water balance model" allows a direct inversion of the RZMC, given parameters describing the canopy interception processes and the vegetation recovery time, and restricting the water balance to only 5 months between May and October, to get rid from the influence of snow (the experimental catchments are located in Oregon and New Hampshire):*
*- Unless vegetation growth is really restricted to these 5 months, this tends to under-estimate the RZMC, and could explain why the Hubbard Brook estimates are so small for forested sites (23 mm on Figure 1)*

We agree with it that vegetation growth is not restricted to these 5 months, but we argue that droughts are restricted to these 5 months. Changing the approach to the full year will indeed result in higher values, but only because water will be stored in the root zone (the simple method does not account for snow), whereas it is actually snow storage. Nevertheless, the actual dry periods are generally in July – August for these catchments. Thus, the deficit of E-P, which actually controls the storage capacity in the root zone, will be the largest in these periods. We would like to clarify here, that for the estimation of the mean Et the full two year period is considered, only the calculation of daily deficits of Et – P was taken over the 5 month summer period.

*- The total evaporation seems to comprise only transpiration and interception loss, and neglect soil evaporation: is it justified?*

It is correct that we do not treat soil evaporation as individual process. Rather, we lump the physical process of evaporation using one interception storage. This will without doubt introduce some uncertainty, but separating the processes is not really warranted by the available data and will result in increased parameter equifinality and thus considerable additional uncertainty. In addition, we argue that our transpiration estimates represent upper limits of transpiration, assuming a negligible amount of soil

evaporation. In reality, the transpiration will indeed be lower due to soil evaporation. We will add a paragraph about this in the discussion.

*- Transpiration depends on a potential evaporation, which is not explained in the paper: does potential evaporation depend on the development of the canopy, as could be quantified by the Leaf Area Index (LAI)? This dependence is well known fact, and can be described for instance by the crop coefficient when following the FAO guidelines of Allen et al. (1986), or as a function of LAI like in the SVAT (Soil-Vegetation-Atmosphere Transfers) models. If such dependence exists in the experimental catchments, it should lead transpiration to decrease after deforestation, and recover with vegetation regrowth, with opposite effects on runoff, in agreement with Figure 2(a-c). In this case, if the model overlooks the positive link between vegetation development and the magnitude of transpiration, it should lead to underestimate the decrease of transpiration after deforestation, and to overestimate the decrease of the RZCM to match the increased observed runoff.*

The potential evaporation was determined based on a temperature based method (Hargreaves equation), and thus did not depend on vegetation. We will add this information in the Methodology. Also, the water balance based model used transpiration estimates, which were exclusively based on the observed water balance. Here, potential evaporation is thus not needed to determine the mean transpiration and was only used to scale the long-term mean value of transpiration to a daily time series.

*- A Monte-Carlo approach is used to assess the effect of the 3 parameters involved in the model (see Table 2) and this allows deriving a very useful uncertainty range around the estimated RZCM. Yet, no justification is given regarding the selected range for these parameters, which is a strong constrain to the uncertainty.*

We would like to refer to Figures S9-S26 in the Supplement. Here, all posterior distributions of the parameters are shown. It can be seen that none of the parameters has an extremely narrow posterior distribution close to one of the bounds of the prior distributions (i.e. upper and lower limits), which would point towards too narrow prior distributions. Only in a few instances, the distributions are close to values of zero, but negative values are not possible for these parameters (e.g. Figure S9b and S9f.) Thus, in general the applied parameter ranges were sufficient for the calibration.

*1.b) The other four models are published conceptual hydrological models, and are calibrated over consecutive 2-yr windows to match the observed water discharge. These models seem to describe the full hydrological year, including the periods of snow, which is a significant difference with the previous approach. Even if some information is given in the Supplementary (but not at the same level for all the models), the reader should find in the main text if the snow is explicitly described, and how the evapotranspiration is calculated (in particular how it depends on the vegetation development, for the same reasons as explained above).*

The conceptual models applied here all use similar functions as originally proposed by Feddes et al. (1978), with the resistance for transpiration as a part of the model (see equations in model descriptions in supplementary material S2). Thus, the models reflect the vegetation influence on transpiration, whereas the potential evaporation exclusively reflects the total energy available for evaporation, which is common practice in the vast majority of hydrological models. All models also used a snow module, as we described in the manuscript (p11,line 12 ; p11, line 27; p12, line 8). Nevertheless, we will try to state more clearly in the model descriptions how evaporation and snow are determined.

*Some details should also be given regarding the calibration itself: How many pa-*

*rameters are calibrated in addition to RZCM (Su,max) for each model? Can all of them change in each 2-yr window, or does only Su,max change? How many tested parameter sets? How many parameter sets are kept at the end of the calibration (equifinality) and what are the corresponding performances to fit the observed discharge? There is a long paragraph from p12L27 to p13L14 which is rather hard to follow for non-specialists of optimization, and could usefully be replaced by objective information regarding the qualities and weakness of the resulting calibration.*

We will add the number of free parameters for calibration in the model descriptions. Generally, almost all parameters were left as free calibration parameters. All parameters in HYMOD (8 parameters) and TUW (15) were free calibration parameters. The 9 parameters of FLEX were all free for calibration, only the slow reservoir coefficient Ks was sampled between narrower bounds, which were based on a recession analysis. The HYPE model used 15 parameters for calibration. We will also add information about the number of initial model runs (100,000 runs) and the number of final feasible parameter sets. The performances for three calibration objective functions (KGE, logKGE and VE) are summarized in Figures S5-S7, for each sub-period of calibration.

*1.c) Another model is used, and presented in 3.5. It's an adaptation of FLEX, one of the above four models, in which an a priori rule for RZCM recovery with time after deforestation is added. First, it would probably be clearer if this model was presented just after the others. Second, much information, again, is lacking:*
*- How is the evolution Imax described since it also varies with time (p15L17-18)?*

We will clarify how Imax changes in time in that model. We applied the same growth function (Equation 11), with growth parameters a and b set to respectively 0.001 [day-1] and 1 [-].

*- How are the parameters a and b of Eq. 11 selected? The resulting values are only given in the caption of Fig8, but don't they deserve some analysis? Do they relate logically to the recovery times that are discussed in section 4.3?*

We will clarify this, but we would also like to refer to lines 12-16 of page 15. The parameters were determined based on a qualitative judgement (thus, just with the 'expert-eye') as it was just meant as a proof-of-concept. We fully acknowledge (p.15, l.20-27) that this is a mere exploratory analysis and a more thorough analysis, which may also include explicit and more detailed process understanding on root development, may be needed to have more adequate values for the growth parameters.

*- How is decided when is RZCM minimum, and which is the minimum value, since Eq. 11 only describes the increasing part of the variations shown on Figure 8?*

The minimum and constant values are determined in the same way as the shape of the curve, with qualitative judgement.

*- Fig 8 shows performance criteria with and without the dynamic formulation of Su,max: to which period do they correspond? We must assume that the period is the full observed period for each catchment, but does it make sense for HB5, where half of the full period is before deforestation? Couldn't it be interesting to test the proposed function over the recovery period only?*

The performance criteria in Fig. 8 correspond to the period just before the treatment until 15 years after the treatment. Therefore, it was not for the full observation period, also for Hubbard Brook WS5. To be more precise, HJ Andrews WS1 was evaluated from 01-10-1960 untill 30-09-1981, Hubbard Brook WS2 from 01-

10-1962 untill 30-09-1983, Hubbard Brook WS5 was evaluated from 01-10-1982 untill 30-09-1999. In this way, we tried to 'zoom in' on the recovery period, just as you suggested, see also page 14, lines 22-25. We will make this clearer in the revision.

*2. The conclusions are too frequently not supported by the Figures. Examples:*
*- p17,L3-4: "the three deforested catchments in the two research forests show generally similar response dynamics after the logging of the catchments (Fig.2)." No, for each of the rows/signatures, you can find one outlier over the three catchments.*

This is why we stated it as 'generally similar response dynamics'. We never claim the responses are exactly the same for all the catchments. We will rephrase this to 'on balance similar response dynamics'.

*- p18, L24-26 (regarding Figure 4): "Comparing the water balance and model-derived estimates of root zone storage capacity SR and Su,max, respectively, then showed that they exhibit very similar patterns in the study catchments." This is abusive since TUW and HYMOD completely miss the difference between HJA and HB, and HB5 doesn't show a clear response to deforestation against inter-annual variability for most models. When discussing Figure 4, the focus is put on the differences in RZCM due to deforestation and recovery. Yet, these differences are much smaller than the ones between the sites, and have a similar magnitude as the inter-annual variability for the two Hubbard Brook catchments. This should be taken in consideration in the discussion.*

We would like to point out that we discuss the pattern, thus the underline{dynamics}, not the absolute values. Especially TUW and HYMOD show a bias (mostly due to the absence of an interception storage) compared with the water-balance method, but still show

similar dynamics (decreasing during deforestation and a gradual increase afterwards). We discussed the possible reasons for the difference between the HJ Andrews and Hubbard Brook catchments (p19, line 5-11 and p20 line 16-18), but we will elaborate more on this in the revision. Briefly, HJ Andrews has a strong seasonal regime, whereas in Hubbard Brook the precipitation is more equally spread throughout the years. Therefore, HJ Andrews has a high need of large root zone storage capacities to allow access to sufficient water throughout the relatively long dry summer period, whereas the Hubbard Brook catchments can survive with much smaller storage volumes, due to significant summer rainfall and thus shorter dry periods that need to be bridged. We agree that inter-annual variability is high, but this is also the reason why we carried out the trend analysis with the undisturbed reference watersheds. In this way, the influence of inter-annual climatic variabilities should be filtered out.

*- p20, L23-26: "It can be argued, that a combination of a relatively long period of low rainfall amounts and high potential evaporation, as can be noted by the relatively high mean annual potential evaporation on top of Figure 4b, led to a high demand in1985". But the top three plots on Fig 4 are so small we can't see much!*

We will make the plots bigger for clarity.

*- p21, L3-4: "Generally, the models applied in Hubbard Brook WS2 show similar behavior as in the HJ Andrews catchment." It's far from being obvious for HB5.*

This is absolutely correct and therefore, we do not state this.

*- p22, L16-17: "The results shown in Figure 4 indicate that these catchments had a rather stable root zone storage capacity during deforestation" (for HJA and HB2).*

*Deforestation is indicated by a red band, and we clearly show a decreasing, not stable, RZCM during deforestation in HJA; for HB2, we don't see anything because the y-axis range is too large.*

We will rephrase this; we basically meant from more or less halfway the period of deforestation (for HJ Andrews just after 1964, and Hubbard Brook WS2 1967). We will try to make the plots clearer as well.

*- p23, L24-28: "Evaluating a set of hydrological signatures suggests that the dynamic formulation of Su,max allows the model to have a higher probability to better reproduce most of the signatures tested here (54% of all signatures in the three catchments) as shown in Figure 9a. A similar pattern is obtained for the more quantitative SRP (Figure 9b), where in 52% of the cases improvements are observed." This is abusive because your get degradation of the performance for 46% of the signatures in Fig9a, and 48% in Fig 9b, which is far from being negligible. If you look at HB5 only, the degraded signatures dominate, which contradicts the conclusion at p24, L27-29.*

We only stated what we found and never deny that 46% and 48% of the signatures show a decrease in performance for the two metrics. Moreover, it is also for these decreasing performances that we added the discussion starting from p24, line 13 until p25, line3, where we explained the origins of these decreases. The statement on p24, line 27-29, also refers to the rather light colors of red and blue, which indicate probabilities around 0.5 and SRP values around 0, thus not a strong preference for one of the two models. We will further clarify this in the revision.

*- p24, L6-7: "In addition, a dynamic formulation of Su,max permits a more plausible representation of the variability in land-atmosphere exchange following land use change". Where does this come from? Provided that no signature in Fig 9 and Table*

*3 addresses the variability of land-atmosphere exchanges (all the signatures describe elements of the streamflow time series).*

We will remove this sentence.

*- p24, L9-10: "Fulfilling its function as a storage reservoir for plant available water, modelled transpiration is significantly reduced post-deforestation, which in turn results in increased runoff coefficients": if I see well on the very small Fig 2c, the results show exactly the opposite for HB5.*

We agree with this, but please note that in the line referred to in this comment to (p24, line9-10) we exclusively discuss the results for HJ Andrews. The two Hubbard Brook catchments are discussed in the following paragraphs.

*- p24, L19-21: "This can also be clearly seen from the hydrographs (Figure 10), where the later part of the recession in the late summer months is much better captured by the time-dynamic model." Personally, I see exactly the opposite, as the time-varying RZCM model in Fig 10b overestimates the peaks, which is not the case of the constant RZCM model in Fig 10a.*

We are confused by this comment, as we clearly see the same considering the peaks in Figure 10b, which we also discuss at page 21, line21-26. We agree that the improvement in the lower parts of the recession (thus not the peaks), is hard to see in Figure 10b, but we still believe this statement is supported by the figure. Please note the additional white space between observed and modelled discharge in the recession of July – August in Figure 10a (time constant model) compared to Figure 10b (time-varying model). To clarify, we will add insets into figs.10a and b, zooming in

to a selected low flow period.

*- Finally, the conclusion relies on a selection of the results that support the assumption of the authors, without considering the results that contradict it, and without a hint of doubt. The limits of the approach (including the model dependency, the small sample of observations which are not perfectly consistent) are not all discussed, nor any alternative frameworks. The authors could for instance consider the possibility that the RZCM could remain unchanged but not fully exploited by the vegetation. This is typically what helps some types of vegetation to resist to drought conditions.*

We tried to keep the discussion brief and stated here the general findings. We believe there are good reasons the results in Hubbard Brook WS5 were less clear, which we also discussed (e.g. p21, line 14 until p22, line 3). Nevertheless, we will add in the discussion and conclusion sections more on several shortcomings and limitations, additional to what we already state in the discussion. We find the remark that root zone storage capacity could remain unchanged very interesting, and we use exactly this argument in our discussion on p19, line 29 until p20, line 6. We will make this clearer in the revision.

**3. Abstract:**
*The abstract is not very clear regarding the methods (the proposed method is not solely based on climate data as written at L8-9, but it requires information on the deforestation, based on inverting the discharge observation in the present case). Like the conclusion, it builds too much on overstatement, but there is also an annoying circular reasoning, since the main conclusion comes from the beginning (L5-7: "Often this parameter [RZCM] is considered to remain constant in time. This is not only conceptually problematic, it is also a potential source of error under the influence of land use and climate change.")*

[Figure]

We will clarify the abstract with the remarks made here. Again, we tried to generalize, which is unfortunately interpreted as an overstatement. Nevertheless, we will add more on the methods and try to clarify.

***4. Other comments:***
*- Trend analysis (method in 3.4, results in 4.3): is it really about trends or about variability? Can we really speak of "trends" on sub-periods as short as those highlighted in blue and green in Fig 7o and 7r? Couldn't these two periods be lumped together? Some references should be given where to find more details on the extraction and interpretation of the 95%-confidence ellipses. Finally, Fig 7 is much too small.*

We agree, at first the method is applied to detect a trend. In the second step, it is used to detect homogeneous sub-periods without a clear trend. We applied the differentiation between sub-periods as objectively as possible, based on the break points in Figures 7d-f. For the construction of the 95%-confidence ellipse, we refer to Equations 9 and 10, and the FAO-guidelines (Allen et al., 1998)

*- Some sentences I did not find clear, although the paper is generally well written:*
*- p3, L13-15: "By extracting plant available water between field capacity and wilting point, roots create moisture storage volumes within their range of influence."*
*- p 4, L7-8: "other species with different water demands may be more in favor in the competition for resources"*
*- p4, L15: "These studies found that deforestation often leads to higher seasonal flows". Do you mean higher peak flows?*
*- p4, 30-31: "More systematic approaches, thus incorporation the change in the model formulation itself"*

*- p14, L28-29: "the calibration was run with a series temporally evolving root zone storage capacities"*
*- p26, L27: I suggest using attributed to rather than caused by, unless a clear causality can be demonstrated.*

We will rephrase the sentences mentioned here.

**References**

Allen, R. G., Pereira, L. S., Raes, D., and Smith, M.: Crop evapotranspiration-Guidelines for computing crop water requirements-FAO Irrigation and drainage paper 56, FAO, Rome, 300, D05109, 1998.

Feddes, R. A., Kowalik, P. J., and Zaradny, H.: Simulation of field water use and crop yield, Centre for Agricultural Publishing and Documentation., 1978.